# Learning Energy Networks
# with Generalized Fenchel-Young Losses

**Mathieu Blondel, Felipe Llinares-López,**
**Robert Dadashi, Léonard Hussenot, Matthieu Geist**
Google Research, Brain team
`{mblondel,fllinares,dadashi,hussenot,mfgeist}@google.com`

## Abstract

Energy-based models, a.k.a. energy networks, perform inference by optimizing an energy function, typically parametrized by a neural network. This allows one to capture potentially complex relationships between inputs and outputs. To learn the parameters of the energy function, the solution to that optimization problem is typically fed into a loss function. The key challenge for training energy networks lies in computing loss gradients, as this typically requires argmin/argmax differentiation. In this paper, building upon a generalized notion of conjugate function, which replaces the usual bilinear pairing with a general energy function, we propose generalized Fenchel-Young losses, a natural loss construction for learning energy networks. Our losses enjoy many desirable properties and their gradients can be computed efficiently without argmin/argmax differentiation. We also prove the calibration of their excess risk in the case of linear-concave energies. We demonstrate our losses on multilabel classification and imitation learning tasks.

## 1 Introduction

Training a neural network usually involves finding its parameters by minimizing a loss function, which captures how well the network fits the data. A typical example is the multiclass logistic loss (a.k.a. cross-entropy loss) from logistic regression, which is the canonical loss associated with the softmax output layer and the categorical distribution. If we replace the softmax with another output layer, what loss function should we use instead? In generalized linear models [52, 48], which include logistic regression and Poisson regression as special cases, the negative log-likelihood gives a loss function associated with a link function, generalizing the softmax to other members of the exponential family [8]. More generally, **Fenchel-Young losses** [18] provide a generic way to construct a canonical convex loss function if the associated output layer can be written in a certain argmax form. Besides the aforementioned generalized linear models, models that fall in this family include sparsemax [47], the structured perceptron [27] and conditional random fields [43, 69]. However, the theory of Fenchel-Young losses is currently limited to argmax output layers that use a **bilinear** pairing.

To increase expressivity, energy-based models [44], a.k.a. **energy networks**, perform inference by optimizing an **energy function**, typically parametrized by a neural network. This leads to an inner optimization problem, which can capture potentially complex relationships between inputs and outputs. A similar approach is taken in SPENs (Structured Prediction Energy Networks) [11, 13], in which a continuous relaxation of the inner optimization problem is used, amenable to projected gradient descent or mirror descent. In input-convex neural networks [4], energy functions are restricted to be convex, so as to make the inner optimization problem easy to solve optimally. To learn the parameters of the energy function, the solution of the inner optimization problem is typically fed into a loss function, leading to an outer optimization problem. In order to solve that problem by stochastic gradient descent, the main challenge lies in computing the loss gradients. Indeed, when

36th Conference on Neural Information Processing Systems (NeurIPS 2022).

using an arbitrary loss function, by the chain rule, computing gradients requires differentiating through the inner optimization problem solution, often referred to as **argmin** or **argmax differentiation**. This can be done by backpropagation through unrolled algorithm iterates [13] or implicit differentiation through the optimality conditions [4]. This issue can be circumvented by using a generalized perceptron loss [44] or a max margin loss [11]. These losses only require **max differentiation**, for which **envelope theorems** [49] can be used. However, these losses often fail to satisfy envelope theorem assumptions such as unicity of the solution and lack theoretical guarantees.

In this paper, we propose to extend the theory of Fenchel-Young losses [18], so as to create a canonical loss function associated with an energy network. Our proposal builds upon **generalized conjugate functions**, also known as **Fenchel-Moreau conjugates** [50, 62], allowing us to go beyond bilinear pairings and to support **general energy functions** such as neural networks. We introduce **regularization**, allowing us to obtain unicity of the solutions and to use envelope theorems to compute gradients, without argmin or argmax differentiation. We provide novel guarantees on the **calibration** of the excess risk, when our loss is used as a surrogate for another discrete loss. To sum up, we obtain a well-motivated loss construction for general energy networks. The rest of the paper is organized as follows.

- After providing some background (§2), we describe **regularized energy networks**, identify a classification of energy functions and give several possible examples that fall in this family (§3).
- We define **generalized conjugates**, an extension of Legendre-Fenchel conjugates, and state their properties (§4). We establish novel conditions for a generalized conjugate to be a smooth function.
- We then introduce **generalized Fenchel-Young losses** and show that they enjoy many of the same favorable properties as the regular Fenchel-Young losses (§5). On the theoretical side, we prove **calibration** of the excess risk for linear-concave energies and strongly-convex regularizers (§6).
- We demonstrate our losses on **multilabel classification** and **imitation learning** tasks (§7).

## 2 Background

**Convex conjugates.** Let $\mathcal{C} \subseteq \mathbb{R}^k$ be an output set. Let $\Omega \colon \mathcal{C} \to \mathbb{R}$ be a function, such that $\Omega(p) = \infty$ for all $p \notin \mathcal{C}$, i.e., $\mathrm{dom}(\Omega) = \mathcal{C}$. Given $v \in \mathcal{V} \subseteq \mathbb{R}^k$, we define the convex conjugate [22] of $\Omega$, also known as the Legendre-Fenchel transform of $\Omega$, by

$$\Omega^*(v) \coloneqq \max_{p \in \mathcal{C}} \ \langle v, p \rangle - \Omega(p). \tag{1}$$

The conjugate $\Omega^*$ is always convex, even if $\Omega$ is not. We define the corresponding argmax as

$$p_\Omega(v) \coloneqq \operatorname*{argmax}_{p \in \mathcal{C}} \ \langle v, p \rangle - \Omega(p) = \nabla \Omega^*(v). \tag{2}$$

The latter equality follows from Danskin's theorem [29, 15], under the assumption that $\Omega$ is strictly convex (otherwise, we obtain a subgradient). It is well-known that $\Omega^*$ is $\frac{1}{\gamma}$-smooth w.r.t. the dual norm $\|\cdot\|_*$ if and only if $\Omega$ is $\gamma$-strongly convex w.r.t. the norm $\|\cdot\|$ [37, 39, 10, 75].

**Fenchel-Young losses.** Suppose that $v = g_\theta(x) \in \mathbb{R}^k$ are the logits / scores produced by a neural network $g$, where $x \in \mathcal{X} \subseteq \mathbb{R}^d$ and $\theta \in \Theta$ are the network's input features and parameters, respectively. What loss function should we use if we want to use (2) as output layer? The Fenchel-Young loss [18] generated by $\Omega \colon \mathcal{C} \to \mathbb{R}$ provides a natural solution. It is defined by

$$L_\Omega(v, y) \coloneqq \Omega^*(v) + \Omega(y) - \langle v, y \rangle, \tag{3}$$

where $y \in \mathcal{Y} \subseteq \mathcal{C}$ is the ground-truth label. Earlier instances of this loss were independently proposed in the contexts of ranking [2] and multiclass classification [30]. Among many useful properties, this loss satisfies $L_\Omega(v, y) \geq 0$ and $L_\Omega(v, y) = 0 \Leftrightarrow y = p_\Omega(v)$ if $\Omega$ is strictly convex [18]. In that sense, it is the canonical loss associated with (2). Interestingly, many existing loss functions are recovered as special cases. For instance, if $\mathcal{C} = \mathbb{R}^k$ and $\Omega(p) = \frac{1}{2}\|p\|_2^2$, which is 1-strongly convex w.r.t. $\|\cdot\|_2$ over $\mathbb{R}^k$, then we obtain the self-dual, the identity mapping and the squared loss:

$$\Omega^*(v) = \frac{1}{2}\|v\|_2^2, \quad p_\Omega(v) = v, \quad L_\Omega(v, y) = \frac{1}{2}\|v - y\|_2^2.$$

If $\mathcal{C}$ is the probability simplex $\triangle^k \coloneqq \{p \in \mathbb{R}_+^k \colon \sum_{i=1}^k p_i = 1\}$ and $\Omega(p)$ is the scaled Shannon negentropy $\gamma\langle p, \log p \rangle$, which is $\gamma$-strongly convex w.r.t. $\|\cdot\|_1$ over $\triangle^k$, then we obtain

$$\Omega^*(v) = \mathrm{LSE}^\gamma(v), \quad p_\Omega(v) = \mathrm{softmax}^\gamma(v), \quad L_\Omega(v, y) = \mathrm{KL}(y, \mathrm{softmax}^\gamma(v)),$$

where we used the log-sum-exp $\text{LSE}^\gamma(v) := \gamma \log(\sum_{i=1}^k \exp(v_i/\gamma))$, $\text{softmax}^\gamma(v) \propto \exp(v/\gamma)$, and the Kullback-Leibler divergence. More generally, when $\mathcal{C} = \text{conv}(\mathcal{Y})$, the convex hull of $\mathcal{Y}$, $p_\Omega$ corresponds to a projection, for which efficient algorithms exist in numerous cases [18, 16]. The calibration of the excess risk of Fenchel-Young losses when $\Omega$ is strongly convex was established in [56, 16]. However, the theory of Fenchel-Young losses is currently limited to the bilinear pairing $\langle v, p \rangle$, which restricts their expressivity and scope.

# 3 Regularized energy networks

**Energy networks.** Also known as energy-based models or EBMs [44], of which SPENS [11, 13] are a particular case, these networks compute predictions by solving an optimization problem of the form

$$p^\Phi(v) := \underset{p \in \mathcal{C}}{\text{argmax}} \; \Phi(v, p),$$

where $v = g_\theta(x) \in \mathcal{V}$ is the energy network's input, $\Phi(v, p)$ is a scalar-valued energy function, and $\mathcal{C}$ is an output set. Throughout this paper, we use the convention that higher energy indicates higher degree of compatibility between $v$ and the prediction $p$. Since $\Phi(v, p)$ is a general energy function, we emphasize that $v$ and $p$ do not need to have the same dimensions, unlike with the bilinear pairing $\langle v, p \rangle$. Any neural network $g_\theta(x) \in \mathbb{R}^k$ can be written in energy network form, since $\text{argmax}_{p \in \mathbb{R}^k} -\|g_\theta(x) - p\|_2^2 = g_\theta(x)$. The key advantage of energy networks, however, is their ability to capture complex interactions between inputs and outputs. This is especially useful in the structured prediction setting [6], where predictions are made of sub-parts, such as sequences being composed of individual elements.

**Introducing regularization.** In this paper, we compute predictions by solving

$$p_\Omega^\Phi(v) := \underset{p \in \mathcal{C}}{\text{argmax}} \; \Phi(v, p) - \Omega(p), \tag{4}$$

where we further added a regularization function $\Omega \colon \mathcal{C} \to \mathbb{R}$. We call $\Phi(v, p) - \Omega(p)$ a **regularized energy function** and we call the corresponding model a **regularized energy network**. We obviously recover usual energy networks by setting $\Omega$ to the indicator function of the set $\mathcal{C}$, i.e., $\Omega(p) = 0$ if $p \in \mathcal{C}$, $\infty$ otherwise. While it is in principle possible to absorb $\Omega$ into $\Phi$, keeping $\Omega$ explicit has several advantages: 1) it allows to introduce generalized conjugate functions and their properties (§4) 2) it allows to introduce generalized Fenchel-Young losses (§5), which mirror the original Fenchel-Young losses 3) we obtain closed forms for (4) in certain cases (Table 1).

**Solving the maximization problem.** The ability to solve the maximization problem in (4) efficiently depends on the properties of $\Phi(v, p)$ w.r.t. $p$. If $\Phi(v, p)$ is **linear** in $p$, for instance $\Phi(v, p) = \langle v, Up \rangle$, then we obtain $p_\Omega^\Phi(v) = p_\Omega(U^\top v)$. Thus, the computation of (4) reduces to (2), for which closed forms are often available. If $\Phi(v, p)$ is **concave** in $p$ and $\mathcal{C}$ is a convex set, then we can solve (4) in polynomial time using an iterative algorithm, such as projected gradient ascent. This is the most general energy class for which our loss and its gradient can be computed to arbitrary precision. If $\Phi(v, p)$ is nonconcave in $p$, then it is not possible to solve (4) in polynomial time in general, unless $\mathcal{C}$ is a discrete set of small cardinality. In general, we emphasize that $\Phi(v, p)$ can be nonconvex in $v$, as is typical with neural networks, since the maximization problem in (4) is with respect to $p \in \mathcal{C}$. However, in the sequel of this paper, certain properties we establish require $\Phi(v, p)$ to be convex in $v$.

We now give examples of regularized energy networks in increasing order of expressivity / complexity.

**Generalized linear models.** As a warm up, we consider the case of **bilinear** energy $\Phi(v, p)$. For instance, for probabilistic classification with $k$ classes, where the goal is to predict $p \in \triangle^k$ from $x \in \mathbb{R}^d$, we can use $\Phi(v, p) = \langle v, p \rangle$ with $v = Wx + b$, where $W \in \mathbb{R}^{k \times d}$ and $b \in \mathbb{R}^k$. In this case, we therefore obtain $p_\Omega^\Phi(v) = p_\Omega(Wx + b)$, recovering generalized linear models [52, 48], the structured perceptron [27] and conditional random fields [43, 69].

**Rectifier and maxout networks.** We now consider the case of **convex-linear** energy $\Phi(v, p)$. A first example is a rectifier network [33] with one hidden layer. Indeed, $\Phi(v, p) = \langle \sigma(v), Up \rangle$ is convex in $v$ if $\sigma$ is an element-wise, convex and non-decreasing activation function, such as the relu or softplus, and if $U$ and $p$ are non-negative. A second example is a maxout network [34] (i.e., a max-affine function) or its smoothed counterpart, the log-sum-exp network [24]. Indeed, $\Phi(v, p) = \sigma(v) \cdot p$, where $\sigma$ is the max or log-sum-exp operator, is convex in $v$, if the scalar $p$ is non-negative. In general, it is always possible to construct a **convex-concave** energy from a jointly-convex function using the Legendre-Fenchel transform in the first argument.

Table 1: Examples of regularized energy networks. The $v$ and $p$ columns indicate the property of the energy function $\Phi(v, p)$ in these variables. The $L_\Omega^\Phi(v, y)$ column indicates the property of the loss in $v$. The linear-quadratic energy uses $v = (A, b)$ where $A$ is negative semi-definite and $\Omega(p) = \frac{\gamma}{2}\|p\|_2^2$.

|  | $\Phi(v, p)$ | $v$ | $p$ | $L_\Omega^\Phi(v, y)$ | $p_\Omega^\Phi(v)$ |
|---|---|---|---|---|---|
| GLM | $\langle v, p \rangle$ | linear | linear | convex | $p_\Omega(v)$ |
| Linear-quadratic | $\frac{1}{2}\langle p, Ap \rangle + \langle p, b \rangle$ | linear | quadratic | convex | $(\gamma I - A)^{-1} b$ |
| Rectifier network | $\langle \mathrm{relu}(v), Up \rangle$ | convex | linear | DC | $p_\Omega(U^\top \mathrm{relu}(v))$ |
| Maxout network | $p \cdot \max(v)$ | convex | linear | DC | $p_\Omega(\max(v))$ |
| LSE network | $p \cdot \mathrm{LSE}^\gamma(v)$ | convex | linear | DC | $p_\Omega(\mathrm{LSE}^\gamma(v))$ |
| ICNN | $-\mathrm{ICNN}(v, p)$ | nonconvex | concave | nonconvex | no closed form |
| Probabilistic | $\sum_{y \in \mathcal{Y}} p(y) E(v, y)$ | nonconvex | linear | nonconvex | $\frac{\exp(E(v, \cdot))}{\sum_{y' \in \mathcal{Y}} \exp(E(v, y'))}$ |
| Arbitrary | $\Phi(v, p)$ | nonconvex | nonconcave | nonconvex | no closed form |

**Input-convex neural networks.** As an example of **nonconvex-concave** energy, we consider $\Phi(v, p) = -\mathrm{ICNN}(v, p)$, where ICNN is an input-convex neural network [4], i.e., it is convex in $p$ but can be nonconvex in $v$. If $\Omega(p)$ and $\mathcal{C}$ are convex, then $\Phi(v, p)$ is concave in $p$ and (4) can be solved in polynomial time using an iterative algorithm, such as projected gradient ascent.

**Probabilistic energy networks.** It is often desirable to define a conditional probability distribution

$$\mathbb{P}(Y = y | X = x) := \frac{\exp(E(v, y))}{\sum_{y' \in \mathcal{Y}} \exp(E(v, y'))}.$$

Such networks are typically trained using a cross-entropy loss (or equivalently, negative log-likelihood). Unfortunately, when $\mathcal{Y}$ is large or infinite, this loss and its gradients are intractable to compute, due to the normalization constant. Therefore, recent research has been devoted to developing approximate training schemes [67, 35, 51]. Although this is not the focus of this paper, we point out that probabilistic energy networks can also be seen as regularized energy networks in the space of probability distributions $\mathcal{C} = \triangle^{|\mathcal{Y}|}$, if we set $\Phi(v, p) = \sum_{y \in \mathcal{Y}} p(y) E(v, y)$ and $\Omega(p) = \sum_{y \in \mathcal{Y}} p(y) \log p(y)$. While the resulting optimization problem is intractable in general, it does suggest possible approximation schemes, such as the use of Frank-Wolfe methods [12, 42, 55].

**Existing loss functions for energy networks.** Given a pair $(x, y)$ and output $v = g_\theta(x)$, how do we measure the discrepancy between $p_\Omega^\Phi(v)$ and $y$? One possibility [4, 13] is to use the composition of a differentiable loss $L \colon \mathcal{C} \times \mathcal{Y} \to \mathbb{R}_+$ with the argmax output, namely $(v, y) \mapsto L(p_\Omega^\Phi(v), y)$. However, computing the loss gradient w.r.t. $v$ then requires to differentiate through $p_\Omega^\Phi(v)$, either through unrolling or implicit differentiation. This is particularly problematic when $\mathcal{C}$ is a complicated convex set, as differentiating through a projection can be challenging. In contrast, our proposed loss completely circumvents this need and enjoys easy-to-compute gradients. For unregularized energy networks, a naive idea, called the energy loss, is to use $(v, y) \mapsto -\Phi(v, y)$. However, this loss only works well if $\Phi$ is a similarity measure and works poorly in general [44]. A better choice is the generalized perceptron loss $(v, y) \mapsto \max_{p \in \mathcal{C}} \Phi(v, p) - \Phi(v, y)$ [44]. Our loss can be seen as a principled generalization of this loss to regularized energy networks, with theoretical guarantees.

## 4 Generalized conjugates

In order to devise generalized Fenchel-Young losses, we build upon a generalization due to Moreau [50, Chapter 14] of the convex conjugate $\Omega^*$. Denoted $\Omega^\Phi$, it replaces the bilinear pairing in (1) with a more general coupling $\Phi$ [62, Chapter 11, Section L]. In this section, we state their definition, properties, closed-form expressions and connection to the $C$-transform in optimal transport.

**Definition.** Let $\Phi(v, p) \in \mathbb{R}$ be a coupling / energy function. The $\Phi$-**convex conjugate** of $\Omega \colon \mathcal{C} \to \mathbb{R}$, also known as **Fenchel-Moreau conjugate**, is then defined by the value function

$$\Omega^\Phi(v) := \max_{p \in \mathcal{C}} \ \Phi(v, p) - \Omega(p). \tag{5}$$

The $\Phi$-convex conjugate is an important tool in abstract convex analysis [66, 63]. Recently, it has been used to provide "Bellman-like" equations in stochastic dynamic programming [25] and to

provide tropical analogues of reproducing kernels [5]. We assume that the maximum is feasible for all $v \in \mathcal{V} = \mathbb{R}^d$, meaning that $\mathrm{dom}(\Omega^\Phi) = \mathcal{V}$. We emphasize again that, unlike with (1), $v$ and $p$ do not need to have compatible dimensions. We denote the argmax solution corresponding to (5) by

$$p_\Omega^\Phi(v) := \underset{p \in \mathcal{C}}{\mathrm{argmax}} \ \Phi(v, p) - \Omega(p). \tag{6}$$

If a function $F(v)$ can be written as $F(v) = \Omega^\Phi(v)$ for some $\Omega$, it is called $\Phi$-**convex** (in analogy, a function $f(v)$ is convex and closed if and only if it can be written as $f(v) = \Omega^*(v)$ for some $\Omega$).

**Properties.** $\Phi$-convex conjugates enjoy many useful properties, some of them are natural extensions of the usual convex conjugate properties. Proofs are provided in Appendix B.1.

---

**Proposition 1** (Properties of $\Phi$-convex conjugates). *Let $\Omega\colon \mathcal{C} \to \mathbb{R}$ and $\Phi\colon \mathcal{V} \times \mathcal{C} \to \mathbb{R}$.*

1. **Generalized Fenchel-Young inequality:** for all $v \in \mathcal{V}$ and $p \in \mathcal{C}$,

$$\Omega^\Phi(v) + \Omega(p) - \Phi(v, p) \geq 0.$$

2. **Convexity:** If $\Phi(v, p)$ is convex in $v$, then $\Omega^\Phi(v)$ is convex (even if $\Omega(p)$ is nonconvex).

3. **Order reversing:** if $\Omega(p) \leq \Lambda(p)$ for all $p \in \mathcal{C}$, then $\Omega^\Phi(v) \geq \Lambda^\Phi(v)$ for all $v \in \mathcal{V}$.

4. **Continuity:** $\Omega^\Phi$ shares the same continuity modulus as $\Phi$.

5. **Gradient (envelope theorem):** Under assumptions (see paragraph below), we have $\nabla \Omega^\Phi(v) = \nabla_1 \Phi(v, p_\Omega^\Phi(v))$, where $\nabla_1$ denotes the gradient in the first argument.

6. **Smoothness:** If $\mathcal{C}$ is a compact convex set, $\Phi(v, p)$ is $\beta$-smooth in $(v, p)$, concave in $p$ and $\Omega(p)$ is $\gamma$-strongly convex in $p$, then $\Omega^\Phi(v)$ is $(\beta + \beta^2/\gamma)$-smooth and $p_\Omega^\Phi(v)$ is $\beta/\gamma$-Lipschitz.

---

The condition on $\Phi$ and $\Omega$ in item 6 for $\Omega^\Phi$ to be a smooth function (i.e., with Lipschitz-continuous gradients) is a novel result and will play a crucial role for establishing calibration guarantees in §6.

**Assumptions for envelope theorems.** The expression in item 5 allows to compute $\nabla \Omega^\Phi(v)$ without *argmax* differentiation. It is based on envelope theorems, which can be used for *max* differentiation. Indeed, we have $\Omega^\Phi(v) = \max_{p \in \mathcal{C}} F(v, p)$, where $F(v, p) := \Phi(v, p) - \Omega(p)$. We assume that the maximum is unique and $\mathcal{C}$ is a compact. If $F(v, p)$ is convex in $v$, we apply Danskin's theorem [29] [15, Proposition B.25]. Without convexity assumption in $v$, if $F(v, p)$ is continuously differentiable in $v$ for all $p \in \mathcal{C}$, $\nabla_1 F$ is continuous, we apply [62, Theorem 10.31]. When we do not compute the exact solution of (6), we only obtain an approximation of the gradient $\nabla \Omega^\Phi(v)$; see [1] for approximation guarantees. For other envelope theorem usecases in machine learning, see, e.g., [21].

**Closed forms.** While (5) and (6) may need to be solved numerically in general, they enjoy closed-form expressions in simple cases. Proofs are provided in Appendix B.2.

---

**Proposition 2** (Closed-form expressions). *Let $\Omega\colon \mathcal{C} \to \mathbb{R}$ and $\Phi\colon \mathcal{V} \times \mathcal{C} \to \mathbb{R}$.*

1. **Bilinear coupling:** If $\Phi(v, p) = \langle v, Up \rangle$, then $\Omega^\Phi(v) = \Omega^*(U^\top v)$ and $p_\Omega^\Phi(v) = p_\Omega(U^\top v)$.

2. **Linear-quadratic coupling:** If $\mathcal{C} = \mathbb{R}^k$, $\Omega(p) = \frac{\gamma}{2}\|p\|_2^2$ and $\Phi(v, p) = \frac{1}{2}\langle p, Ap \rangle + \langle p, b \rangle$, where $v = (A, b)$ and $A$ is such that $(\gamma I - A)$ is positive definite, we obtain

$$\Omega^\Phi(v) = \frac{1}{2}\langle b, (\gamma I - A)^{-1}b \rangle \quad \text{and} \quad p_\Omega^\Phi(v) = (\gamma I - A)^{-1}b. \tag{7}$$

3. **Metric coupling:** If $\mathcal{V} = \mathcal{C}$, $\Phi = -C$ where $C(v, p)$ is a metric and $\Omega$ is $M$-Lipschitz with $M \leq 1$, then $\Omega^\Phi = -\Omega$ and $p_\Omega^\Phi(v) = v$.

---

**Relation with the $C$-transform.** Given a cost function $C$, we may define a $\min$ counterpart of (5),

$$\Lambda_C(v) := \min_{p \in \mathcal{C}} \ C(v, p) - \Lambda(p). \tag{8}$$

In the optimal transport literature, this is known as the $C$-transform of $\Lambda$ [65, 60]. When $C$ is bilinear, this recovers the notion of concave conjugate [20]. When $C(v, p) = c(v - p)$ for some $c$, this recovers the infimal convolution [20]. It is easy to check that $\Omega = -\Lambda \Leftrightarrow \Omega^\Phi = -\Lambda_C$ with $C = -\Phi$. Thus, $\Omega^\Phi$ and $\Lambda_C$ are the natural extensions of convex and concave conjugates, respectively. We opt for the former in this paper to closely mirror the usual convex conjugates and Fenchel-Young losses.

# 5 Generalized Fenchel-Young losses

**Definition.** The generalized Fenchel-Young inequality in Proposition 1 leads us to propose the generalized Fenchel-Young loss, the natural extension of (3) to $\Phi$-convex conjugates:

$$L_\Omega^\Phi(v, y) \coloneqq \Omega^\Phi(v) + \Omega(y) - \Phi(v, y). \tag{9}$$

We also have the relationships $L_\Omega^\Phi(v, y) = F(v, y) - F(v, p_\Omega^\Phi(v))$, where $F(v, p) = \Omega(p) - \Phi(v, p)$, and $p_\Omega^\Phi(v) = \operatorname{argmin}_{p \in \mathcal{C}} L_\Omega^\Phi(v, p)$. We now give an intuitive geometric interpretation of (9).

**Geometric interpretation.** For the sake of illustration, let us set $\Omega(p) = \frac{\gamma}{2}p^2$ and $\Phi(v, p) = \frac{1}{2}ap^2 + bp$. From (1), the line $p \mapsto \langle u, p \rangle - \Omega^*(u)$, where $\Omega^*(u) = \frac{1}{2\gamma}u^2$, is the tightest linear lower bound on $\Omega(p)$, here depicted with $u = 0.6$. It is the tangent of $\Omega(p)$ at $p = p_\Omega(u)$. Similarly, from (5), $p \mapsto \Phi(v, p) - \Omega^\Phi(v)$ is the tightest lower-bound of this form given $v$. It touches $\Omega(p)$ at $p = p_\Omega^\Phi(v)$, here depicted with $v = (a, b)$, $a = -1$ and $b = 1$, for which we have the closed form $\Omega^\Phi(v) = \frac{1}{2}(\gamma - a)^{-1}b^2$ (Proposition 2). The generalized Fenchel-Young loss (9) is then the **gap** between $\Omega(p)$ and $\Phi(v, p) - \Omega^\Phi(v)$, evaluated at the ground-

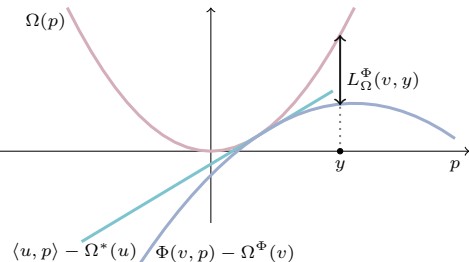

Figure 1: Geometric interpretation with $\Omega(p) = \frac{\gamma}{2}p^2$ and $\Phi(v, p) = \frac{1}{2}ap^2 + bp$.

truth $p = y$. The goal of training is to adjust the parameters $\theta$ of a network $v = g_\theta(x)$ so as to minimize this gap averaged over all $(x, y)$ training pairs.

**Properties.** Generalized Fenchel-Young losses enjoy many desirable properties, as we now show.

---

**Proposition 3** (Properties of generalized F-Y losses). *Let $\Omega \colon \mathcal{C} \to \mathbb{R}$ and $\Phi \colon \mathcal{V} \times \mathcal{C} \to \mathbb{R}$.*

1. **Non-negativity:** $L_\Omega^\Phi(v, p) \geq 0$ for all $v \in \mathcal{V}$ and $p \in \mathcal{C}$.

2. **Zero loss:** if the maximum in (5) exists and is unique, $L_\Omega^\Phi(v, y) = 0 \Leftrightarrow y = p_\Omega^\Phi(v)$.

3. **Gradient:** $\nabla_1 L_\Omega^\Phi(v, p) = \nabla \Omega^\Phi(v) - \nabla_1 \Phi(v, p)$.

4. **Difference of convex (DC):** if $\Phi(v, p)$ is convex in $v$, then $L_\Omega^\Phi(v, p)$ is a difference of convex functions in $v$: $\Omega^\Phi(v)$ and $\Phi(v, p)$. If $\Phi(v, p)$ is linear in $v$, then $L_\Omega^\Phi(v, p)$ is convex in $v$.

5. **Smaller output set, smaller loss.** If $\mathcal{C}' \subseteq \mathcal{C}$ and $\Omega'$ is the restriction of $\Omega$ to $\mathcal{C}'$, then $L_{\Omega'}^\Phi(v, p) \leq L_\Omega^\Phi(v, p)$ for all $v \in \mathcal{V}$ and $p \in \mathcal{C}'$.

6. **Quadratic lower-bound.** If $\Phi(v, p) - \Omega(p)$ is $\gamma$-strongly concave in $p$ w.r.t. $\|\cdot\|$ over $\mathcal{C}$ and $\mathcal{C}$ is a closed convex set, then for all $v \in \mathcal{V}$ and $p \in \mathcal{C}$

$$\frac{\gamma}{2}\|p - p_\Omega^\Phi(v)\|^2 \leq L_\Omega^\Phi(v, p). \tag{10}$$

7. **Upper-bounds.** If $\Phi(v, p) - \Omega(p)$ is $\alpha$-Lipschitz in $p$ w.r.t. $\|\cdot\|$ over $\mathcal{C}$, then $L_\Omega^\Phi(v, p) \leq \alpha\|p - p_\Omega^\Phi(v)\|$. If $\Phi(v, p)$ is concave in $p$, then $L_\Omega^\Phi(v, p) \leq L_\Omega(\nabla_2\Phi(v, p), p)$.

---

Proofs are given in Appendix B.3. The **non-negativity** is a good property for a loss function. The **zero-loss** property $L_\Omega^\Phi(v, y) = 0 \Leftrightarrow y = p_\Omega^\Phi(v)$, which is true for instance if $p \mapsto \Phi(v, p) - \Omega(p)$ is strictly concave, is key as it allows to use $p_\Omega^\Phi(v)$ defined in (6) as the (implicit) output layer associated with an energy network. The **gradient** $\nabla_1 L_\Omega^\Phi(v, y)$ does not require to differentiate through the argmax problem in (6) needed for computing $p_\Omega^\Phi(v)$. Typically, differentiating through an argmax or argmin, as is done in input-convex neural networks [4], requires either unrolling or implicit differentiation [36, 14, 41, 19, 17] and is therefore more costly.

The fact that $L_\Omega^\Phi(v, p)$ is a **difference of convex (DC)** functions in $v$ when $\Phi(v, p)$ is convex in $v$ suggests that we can use DC programming techniques, such as the convex-concave procedure [72], for training such energy networks. We leave the investigation of this observation to future work. The **"smaller output set, smaller loss"** property means that we can achieve the smallest loss by choosing the smallest set $\mathcal{C}$ in (6) such that $\mathcal{Y} \subseteq \mathcal{C}$. The smallest such convex set is the convex hull of $\mathcal{Y}$, also

known as marginal polytope [71] when $\mathcal{Y} \subseteq \{0,1\}^k$. The **quadratic lower-bound** relates our loss to using $p_\Omega^\Phi(v)$ within a squared norm loss. The **upper-bounds** relate our loss to using $p_\Omega^\Phi(v)$ within a norm loss and to using a regular Fenchel-Young loss with a linearized energy.

If $\Omega$ is the indicator function of $\mathcal{C}$, i.e., $\Omega(p) = 0$ if $p \in \mathcal{C}$, $\infty$ otherwise, then it can be checked that we recover the "generalized perceptron" loss [45, 44] as a special case of (9). Proposition 3 therefore provides new properties to understand and analyze this loss.

Regular Fenchel-Young losses are closely related to Bregman divergences [18]. In Appendix A, we build a generalized notion of Bregman divergence using generalized conjugates.

**Training.** To train regularized energy networks with our framework, the user should choose an energy $\Phi(v, p)$, a regularization $\Omega(p)$, an output set $\mathcal{C}$ (from Proposition 3 item 5, the smaller this set the better) and the model $v = g_\theta(x)$ with input $x$ and parameters $\theta$. Given a set of input-output pairs $(x_1, y_1), \ldots, (x_n, y_n) \in \mathcal{X} \times \mathcal{Y}$, where $\mathcal{Y} \subseteq \mathcal{C}$, we can find the parameters $\theta$ by minimizing the empirical risk objective regularized by $R: \Theta \to \mathbb{R}$,

$$\widehat{\theta} = \underset{\theta \in \Theta}{\operatorname{argmin}} \frac{1}{n} \sum_{i=1}^{n} L_\Omega^\Phi(g_\theta(x_i), y_i) + R(\theta). \tag{11}$$

Thanks to the easy-to-compute gradients of $L_\Omega^\Phi$, we can easily solve (11) using any (stochastic) solver.

## 6   Calibration guarantees

Many times, notably for differentiability reasons, the loss used at training time, here our generalized Fenchel-Young loss $L_\Omega^\Phi(v, y)$, is used as a surrogate / proxy for a different (potentially discrete loss) $L: \mathcal{Y} \times \mathcal{Y} \to \mathbb{R}$, used at test time. Calibration guarantees [73, 74, 9, 68] ensure that minimizing the excess of risk of the train loss will also minimize that of the test loss (a.k.a. target loss). We study in this section such guarantees, assuming that $L$ satisfies an affine decomposition property [26, 16]:

$$L(\widehat{y}, y) = \langle \varphi(\widehat{y}), V\varphi(y) + b \rangle + c(y), \tag{12}$$

where $\varphi(y) \mapsto V\varphi(y) + b$ is an affine map, $\varphi(y)$ is a label embedding and $c(y)$ is any function that depends only on $y$. Numerous losses can be written in this form. Examples include the zero-one, Hamming, NDCG and precision at $k$ losses [56, 16]. Inference in this setting works in two steps. First, we compute a "soft" (continuous) prediction $p = p_\Omega^\Phi(v) \in \mathcal{C}$. Second, we compute a "hard" (discrete) prediction by a decoding / rounding from $\mathcal{C}$ to $\mathcal{Y}$, calibrated for the loss $L$:

$$y_L(p) := \underset{\widehat{y} \in \mathcal{Y}}{\operatorname{argmin}} \, L(\widehat{y}, p) = \underset{\widehat{y} \in \mathcal{Y}}{\operatorname{argmin}} \langle \varphi(\widehat{y}), V\varphi(p) + b \rangle. \tag{13}$$

The target risk of $f: \mathcal{X} \to \mathcal{Y}$ and the surrogate risk of $g: \mathcal{X} \to \mathcal{V}$ are defined by

$$\mathcal{L}(f) := \mathbb{E}_{(X,Y) \sim \rho} \, L(f(X), Y) \quad \text{and} \quad \mathcal{L}_\Omega^\Phi(g) := \mathbb{E}_{(X,Y) \sim \rho} \, L_\Omega^\Phi(g(X), Y),$$

where $\rho$ is a typically unknown distribution over $\mathcal{X} \times \mathcal{Y}$. The Bayes predictors are defined by $f^\star := \operatorname{argmin}_{f: \mathcal{X} \to \mathcal{Y}} \mathcal{L}(f)$ and $g^\star := \operatorname{argmin}_{g: \mathcal{X} \to \mathcal{V}} \mathcal{L}_\Omega^\Phi(g)$. We now establish calibration of the surrogate excess risk, under the assumption that the energy $\Phi(v, p)$ is **linear-concave**, i.e., it can be written as $\Phi(v, p) = \langle v, \varphi(p) \rangle$, for some function $\varphi$.

---

**Proposition 4.** *Calibration of target and surrogate excess risks*

*Assume $L_\Omega^\Phi(v, y)$ is $M$-smooth in $v$ w.r.t. the dual norm $\|\cdot\|_*$, $\mathcal{C}$ is a compact convex set such that $\mathcal{Y} \subseteq \mathcal{C}$ and $\Phi(v, p) = \langle v, \varphi(p) \rangle$. Let $\sigma := \sup_{y \in \mathcal{Y}} \|V^\top \varphi(y)\|_*$. Then, the generalized Fenchel-Young loss (9) is calibrated with the target loss (12) with decoder $d = y_L \circ p_\Omega^\Phi$:*

$$\forall g: \mathcal{X} \to \mathcal{V} \quad \frac{(\mathcal{L}(d \circ g) - \mathcal{L}(f^\star))^2}{8\sigma^2 M} \leq \mathcal{L}_\Omega^\Phi(g) - \mathcal{L}_\Omega^\Phi(g^\star),$$

*where $p_\Omega^\Phi: \mathcal{V} \to \mathcal{C}$ is defined in (6) and $y_L: \mathcal{C} \to \mathcal{Y}$ is defined in (13).*

---

The proof is in Appendix B.4. Proposition 1 item 6 shows that the smoothness of $\Phi$ and strong convexity of $\Omega$ ensure the smoothness of $L_\Omega^\Phi$. Calibration also implies Fisher consistency, namely

Table 2: Multilabel classification results using various energies (test accuracy in %).

| Energy | yeast | scene | mediamill | birds | emotions | cal500 |
|---|---|---|---|---|---|---|
| Unary (linear) | 79.76 | 89.14 | 96.84 | 86.47 | 78.22 | 85.67 |
| Unary (rectifier network) | 80.03 | 91.35 | 96.91 | 91.74 | 79.79 | 86.25 |
| Pairwise | **80.19** | **91.58** | **96.95** | 91.55 | **80.56** | 85.73 |
| SPEN | 79.99 | 91.24 | 96.68 | 91.41 | 79.35 | 86.25 |
| Input-concave SPEN | 80.00 | 90.64 | **96.95** | **91.77** | 79.73 | **86.35** |

$\mathcal{L}_\Omega^\Phi(g) = \mathcal{L}_\Omega^\Phi(g^\star) \Rightarrow \mathcal{L}(d \circ g) = \mathcal{L}(f^\star)$, when using the decoder $d = y_L \circ p_\Omega^\Phi$. The existing proof technique for the calibration of regular Fenchel-Young losses [56, 16] assumes a bilinear pairing and a loss of the form $L_{\Omega_\varphi}(u, \varphi(y)) = \Omega_\varphi^*(u) + \Omega_\varphi(\varphi(y)) - \langle u, \varphi(y) \rangle$, where $\Omega_\varphi$ is a strongly-convex regularizer w.r.t. $\varphi(y)$. Our novel proof technique is more general, as it works with any linear-concave energy and the regularizer $\Omega$ is w.r.t. $y$, not $\varphi(y)$. Unlike the existing proof, our proof is valid for the pairwise model we present in the next section.

## 7 Experiments

### 7.1 Multilabel classification

We study in this section the application of generalized Fenchel-Young losses to multi-label classification, setting $\mathcal{Y} = \{0, 1\}^k$ and $\mathcal{C} = [0, 1]^k$, where $k$ is the number of labels. When the loss $L$ in (12) is the Hamming loss (1 - accuracy), our loss is calibrated for $L$ and the decoding (13) is just $\widehat{y}_j = 1$ if $p_j > 0.5$ else 0. We therefore report our empirical results using the accuracy metric.

**Unary model.** We consider a neural network $u = g_\theta(x) \in \mathbb{R}^k$, assigning a score $u_j$ to each label $j \in [k]$. With the bilinear pairing $\Phi(u, p) = \langle u, p \rangle$, we get

$$p_\Omega^\Phi(u) = \underset{p \in [0,1]^k}{\mathrm{argmax}} \langle u, p \rangle - \Omega(p) = p_\Omega(u). \tag{14}$$

That is, (14) is just a normal neural network with $p_\Omega$ as output layer. When $\Omega(p) = \Omega_1(p) + \Omega_1(1-p)$, where $\Omega_1(p) := \langle p, \log p \rangle$ is Shannon's negentropy, we get $p_\Omega(v) = \mathrm{sigmoid}(v) := 1/(1 + \exp(-v))$ and (9) is just the usual binary logistic / cross-entropy loss. When $\Omega(p) = \Omega_2(p) + \Omega_2(1-p)$, where $\Omega_2(p) := \frac{1}{2}\langle p, 1-p \rangle$ is Gini's negentropy, we get a sparse sigmoid and the binary sparsemax loss [18, §6.2]. Because $\mathcal{C} = [0, 1]^k$ is the convex hull of $\mathcal{Y} = \{0, 1\}^k$, (14) can be intepreted as a **marginal probability** [71]. Indeed, there exists a probability distribution $\mathbb{P}(Y|X)$ over $Y \in \mathcal{Y}$ such that

$$[p_\Omega^\Phi(u)]_j = \mathbb{P}(Y_j = 1 | X = x).$$

**Pairwise model.** We now additionally use a network $U = h_\theta(x) \in \mathbb{R}^{k \times k}$, assigning a score $U_{i,j}$ to the pairwise interaction between labels $i$ and $j$. With $v = (u, U)$ and the **linear-quadratic** coupling $\Phi(v, p) = \langle u, p \rangle + \frac{1}{2}\langle p, Up \rangle$, we get

$$p_\Omega^\Phi(v) = \underset{p \in [0,1]^k}{\mathrm{argmax}} \langle u, p \rangle + \frac{1}{2}\langle p, Up \rangle - \Omega(p). \tag{15}$$

If $U$ is negative semi-definite, i.e., $U = -AA^\top$ for some matrix $A \in \mathbb{R}^{k \times m}$, then the problem is concave in $p$ and can be solved optimally. Moreover, since $\Phi(v, p)$ is linear-concave, the calibration guarantees in §6 hold and $L_\Omega^\Phi(v, y)$ is convex in $v$. In our experiments, we use a matrix $A$ of rank 1 (cf. Appendix C). Unlike (7), (15) does not enjoy a closed form due to the constraints. We solve it by coordinate ascent: if $\Omega$ is quadratic, as is the case with Gini's negentropy, the coordinate-wise updates can be computed in closed-form. Again, (15) can be interpreted as a marginal probability. In constrast to (15), marginal inference in the closely related **Ising model** is known to be #P-hard [32].

**SPEN model.** Following SPENs [11, Eq. 4 and 5], we also tried the energy $\Phi(v, p) = \langle u, p \rangle - \Psi(w, p)$, where $v = (u, w)$, $u = g_\theta(x)$ and $w$ are the weights of the "prior network" $\Psi$ (independent of $x$). We also tried a variant where $\Psi$ is made convex in $p$, making $\Phi(v, p)$ concave in $p$. In both cases, we compute $p_\Omega^\Phi(v)$ by solving (6) using projected gradient ascent with backtracking linesearch.

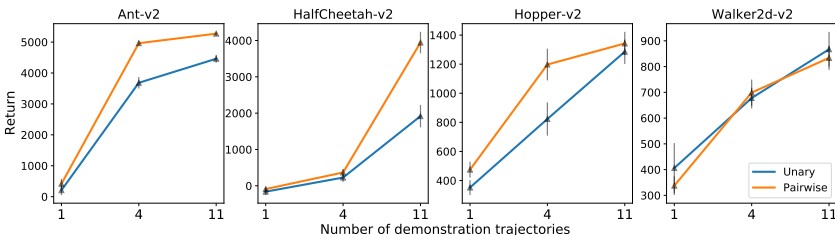

Figure 2: Average performance (higher is better) and standard deviation over 10 seeds.

**Experimental setup.** We perform experiments on 6 publicly-available datasets, see Appendix C. We use the train-test split from the dataset when provided. When not, we use 80% for training data and 20% for test data. For all models, we solve the outer problem (11) using ADAM. We set $\Omega(p)$ to the Gini negentropy. We hold out 25% of the training data for hyperparameter validation purposes. We set $R(\theta)$ in (11) to $\frac{\lambda}{2}\|\theta\|_2^2$. For the regularization hyper-parameter $\lambda$, we search 5 log-spaced values between $10^{-4}$ and $10^1$. For the learning rate parameter of ADAM, we search 10 log-spaced valued between $10^{-5}$ and $10^{-1}$. Once we selected the best hyperparameters, we refit the model on the entire training set. We average results over 3 runs with different seeds.

**Results.** Table 2 shows a model comparison. We observe improvements with the pairwise model on 4 out of 6 datasets, confirming that our losses are able to learn useful models. Input-concavity helps improve SPENs in 4 out of 6 datasets. Table 4 confirms that using the envelope theorem for computing gradients works comparably to (if not better than) the implicit function theorem. Table 5 shows that our losses outperform the energy, the cross-entropy and the generalized perceptron losses.

### 7.2 Imitation learning

In this section, we study the application of generalized Fenchel-Young losses to imitation learning. This setting consists in learning a policy $\pi : \mathcal{X} \mapsto \mathcal{Y}$, a mapping from states $\mathcal{X}$ to actions $\mathcal{Y}$, from a fixed dataset of expert demonstrations $(x_1, y_1), \ldots, (x_n, y_n) \in \mathcal{X} \times \mathcal{Y}$. In particular, we only consider a Behavior Cloning approach [61], which essentially reduces imitation learning to supervised learning (as opposed to inverse RL methods [64, 54]). The learned policy $\pi$ is evaluated on its performance, which is the expected sum of rewards of the environment [70].

**Experimental setup.** We consider four MuJoCo Gym locomotion environments [23] together with the demonstrations provided by Orsini et al. [58]; see Appendix C.2 for details. We evaluate the learned policy $\pi$ for different number of demonstration trajectories: 1, 4 and 11, consistently with [38, 40, 28]. The action space in the demonstrations is included in $[-1, 1]^k$, where the dimensionality of the action space $k$ corresponds to the torques of the actuators. We scale the action space to the hypercube $\mathcal{Y} = \mathcal{C} = [0, 1]^k$ at learning time and scale it back to the original action space at inference time. Similarly to the multilabel classification setup, we evaluate the unary and pairwise models, with the only difference that we use two hidden layers instead of one, since it leads to significantly better performance. We specify the hyperparameter selection procedure in Appendix C.2.

**Results.** We run the best hyperparameters over 10 seeds and report final performance over 100 evaluation episodes. Figure 2 shows a clear improvement of the pairwise model over the unary model for 3 out of 4 tasks. Contrary to the unary model, the pairwise model enables to capture the interdependence between the different torques of the action space, translating into better performance.

## 8 Conclusion

Building upon generalized conjugate functions, we proposed generalized Fenchel-Young losses, a natural loss construction for learning energy networks and studied its properties. Thanks to conditions on the energy $\Phi$ and the regularizer $\Omega$ ensuring the loss smoothness, we established calibration guarantees for the case of linear-concave energies, a more general result than the existing analysis, restricted to bilinear energies. We demonstrated the effectiveness of our losses on multilabel classification and imitation learning tasks. We hope that this paper will help popularize generalized conjugates as a powerful tool for machine learning and optimization.

## Acknowledgments

MB thanks Gabriel Peyré for numerous discussions on $C$-transforms and Clarice Poon for discussions on envelope theorems, as well as Vlad Niculae and André Martins for many fruitful discussions.

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
