# A   Generalized Bregman divergences

The expression $\Omega^*(v) + \Omega(p) - \langle v, p \rangle$ at the heart of Fenchel-Young losses is closely related to Bregman divergences; see [3, Theorem 1.1] and [18]. In this section, we develop a similar relationship between generalized Fenchel-Young losses and a new generalized notion of Bregman divergence.

**Biconjugates.**   We begin by recalling well-known results on biconjugates [22]. Applying the conjugate (1) twice, we obtain the biconjugate $\Omega^{**}(p)$. It is well-known that a function $\Omega$ is convex *and* closed (i.e., lower-semicontinuous) if and only if $\Omega = \Omega^{**}$. This therefore provides a variational characterization of lower-semicontinuous convex functions. This characterization naturally motivates the class of $\Phi$-convex functions (§4). If $\Omega$ is nonconvex, $\Omega^{**}$ is $\Omega$'s tightest convex lower bound.

**Generalized biconjugates.**   The generalized conjugate in (5) uses maximization w.r.t. the second argument $p \in \mathcal{C}$. To obtain a generalized conjugate whose maximization is w.r.t. the first argument $v \in \mathcal{V}$ instead, we define $\Psi(p, v) \coloneqq \Phi(v, p)$. Note that if $\Phi$ is symmetric, the distinction between $\Phi$ and $\Psi$ is not necessary. Similarly to (5), we can then define the $\Psi$-conjugate of $\Lambda \colon \mathcal{V} \to \mathbb{R}$ as

$$\Lambda^{\Psi}(p) = \max_{v \in \mathcal{V}} \ \Psi(p, v) - \Lambda(v) = \max_{v \in \mathcal{V}} \ \Phi(v, p) - \Lambda(v), \tag{16}$$

i.e., the maximization is over the left argument of $\Phi$. We define the corresponding argmax as

$$v_{\Lambda}^{\Psi}(p) \coloneqq \operatorname*{argmax}_{v \in \mathcal{V}} \ \Psi(p, v) - \Lambda(v) = \operatorname*{argmax}_{v \in \mathcal{V}} \ \Phi(v, p) - \Lambda(v). \tag{17}$$

In particular, with $\Lambda \coloneqq \Omega^{\Phi}$, we can define the generalized biconjugate $\Lambda^{\Psi} = \Omega^{\Phi\Psi} \colon \mathcal{C} \to \mathbb{R}$. Generalized biconjugates enjoy similar properties as regular biconjugates, as we now show.

---

**Proposition 5** (Properties of generalized biconjugates). *Let* $\Omega \colon \mathcal{C} \to \mathbb{R}$*,* $\Phi \colon \mathcal{V} \times \mathcal{C} \to \mathbb{R}$ *and* $\Psi(p, v) \coloneqq \Phi(v, p)$.

1. **Lower-bound:** $\Omega^{\Phi\Psi}(p) \le \Omega(p)$ for all $p \in \mathcal{C}$.

2. **Equality:** $\Omega^{\Phi\Psi}(p) = \Omega(p)$ if and only if $\Omega$ is $\Psi$-convex.

3. **Tightest lower-bound:** $\Omega^{\Phi\Psi}(p)$ is the tightest $\Psi$-convex lower-bound of $\Omega(p)$.

---

Proofs are given in Appendix B.5. Similar results hold for $\Lambda^{\Psi\Phi}$, where $\Lambda \colon \mathcal{V} \to \mathbb{R}$.

**Definition.**   We can now define the **generalized Bregman divergence** $D_{\Omega}^{\Phi} \colon \mathcal{C} \times \mathcal{C} \to \mathbb{R}_+$ as

$$D_{\Omega}^{\Phi}(p, p') \coloneqq \Omega(p) - \Phi(v_{\Omega^{\Phi}}^{\Psi}(p'), p) - \Omega(p') + \Phi(v_{\Omega^{\Phi}}^{\Psi}(p'), p'), \tag{18}$$

where we recall that $\Psi(p, v) \coloneqq \Phi(v, p)$ and using (17) we have

$$v_{\Omega^{\Phi}}^{\Psi}(p) = \operatorname*{argmax}_{v \in \mathcal{V}} \ \Psi(p, v) - \Omega^{\Phi}(v) = \operatorname*{argmax}_{v \in \mathcal{V}} \ \Phi(v, p) - \Omega^{\Phi}(v). \tag{19}$$

We have therefore obtained a notion of Bregman divergence parametrized by a coupling $\Phi(v, p)$, such as a neural network.

As shown in the proposition below, generalized Bregman divergences enjoy similar properties as regular Bregman divergences. Proofs are given in Appendix B.6.

> **Proposition 6** (Properties of generalized Bregman divergences)**.** *Let $\Omega$ be a $\Psi$-convex function, where $\Psi(p, v) := \Phi(v, p)$.*
>
> 1. **Link with generalized FY loss:** Denoting $v := v_{\Omega^\Phi}^\Psi(p')$, we have $D_\Omega^\Phi(p, p') = L_\Omega^\Phi(v, p)$.
>
> 2. **Non-negativity:** $D_\Omega^\Phi(p, p') \geq 0$, for all $p, p' \in \mathcal{C}$.
>
> 3. **Identity of indiscernibles:** $D_\Omega^\Phi(p, p') = 0 \Leftrightarrow p = p'$ if $p \mapsto \Phi(v, p) - \Omega(p)$ is strictly concave for all $v \in \mathcal{V}$.
>
> 4. **Convexity:** if $p \mapsto \Phi(v, p) - \Omega(p)$ is concave for all $v \in \mathcal{V}$, then $D_\Omega^\Phi(p, p')$ is convex in $p$.
>
> 5. **Recovering Bregman divergences:** If $\Phi(v, p)$ is the bilinear coupling $\langle v, p \rangle$, then we recover the usual Bregman divergence $D_\Omega : \mathcal{C} \times \mathcal{C} \to \mathbb{R}_+$
>
> $$D_\Omega^\Phi(p, p') = D_\Omega(p, p') := \Omega(p) - \Omega(p') - \langle \nabla\Omega(p'), p - p' \rangle. \tag{20}$$

Some remarks:

- The generalized Bregman divergence is between objects $p$ and $p'$ of the same space $\mathcal{C}$, while the generalized Fenchel-Young loss is between objects $v$ and $p$ of mixed spaces $\mathcal{V}$ and $\mathcal{C}$.

- If $\Phi(v, p) - \Omega(p)$ is concave in $p$, then $D_\Omega^\Phi(p, p')$ is convex in $p$, as is the case of the usual Bregman divergence $D_\Omega(p, p')$. However, (19) is not easy to solve globally in general, as it is the maximum of a difference of convex functions in $v$. This can be done approximately by using the convex-concave procedure [72], linearizing the left part. We have the opposite situation with the generalized Fenchel-Young loss: if $\Phi(v, p) - \Omega(p)$ is convex-concave, $L_\Omega^\Phi(v, y)$ is easy to compute but it is a difference of convex functions in $v$.

- Similarly, we may also define $D_{\Omega^\Phi}^\Psi(v, v') = \Omega^\Phi(v) - \Psi(p_\Omega^\Phi(v'), v) - \Omega^\Phi(v') + \Psi(p_\Omega^\Phi(v'), v')$, where we recall that $p_\Omega^\Phi(v) := \mathrm{argmax}_{p \in \mathcal{C}} \, \Phi(v, p) - \Omega(p)$. We have thus obtained a divergence between two objects $v$ and $v'$ in the same space $\mathcal{V}$.

# B   Proofs

## B.1   Proofs for Proposition 1 (properties of generalized conjugates)

1. **Generalized Fenchel-Young inequality.** From (5), we immediately obtain $\Omega^\Phi(v) \geq \Phi(v, p) - \Omega(p)$ for all $v \in \mathcal{V}$ and all $p \in \mathcal{C}$.

2. **Convexity.** If $\Phi(v, p)$ is convex in $v$, then $\Omega^\Phi(v)$ is the maximum of a family of convex functions indexed by $p$. Therefore, $\Omega^\Phi(v)$ is convex. Note that this is the case even if $\Omega(p)$ is nonconvex.

3. **Order reversing.** Since $\Omega \leq \Lambda$, we have

$$\Omega^\Phi(v) = \max_{p \in \mathcal{C}} \Phi(v, p) - \Omega(p) \geq \max_{p \in \mathcal{C}} \Phi(v, p) - \Lambda(p) = \Lambda^\Phi(v).$$

4. **Continuity.** See [65, Box 1.8].

5. **Gradient.** See "Assumptions for envelope theorems" in §4.

6. **Smoothness.** We follow the proof technique of [46, Lemma 4.3], which states that $v \mapsto \max_{p \in \mathcal{C}} E(v, p)$ is $(\beta + \beta^2/\gamma)$-smooth if $E$ is $\beta$-smooth in $(v, p)$ and $\gamma$-strongly concave in $p$ over $\mathcal{C}$. We show here that if $E(v, p)$ decomposes as $E(v, p) = \Phi(v, p) - \Omega(p)$, where $\Omega$ is $\gamma$-strongly convex over $\mathcal{C}$ **but possibly nonsmooth** (as is the case for instance of Shannon's negentropy), and $\Phi(v, p)$ is $\beta$-smooth in $(v, p)$ and concave in $p$, then we still have that $\Omega^\Phi(v) = \max_{p \in \mathcal{C}} \Phi(v, p) - \Omega(p)$ is $(\beta + \beta^2/\gamma)$-smooth.

   For brevity, let us define the shorthand $p_\Omega^\Phi(v) := p^\star(v)$. From [46, Section A.3], we have

$$\gamma\|p^\star(v_2) - p^\star(v_1)\|^2 \leq \langle p^\star(v_2) - p^\star(v_1), \nabla_2 E(v_2, p^\star(v_2)) - \nabla_2 E(v_1, p^\star(v_2)) \rangle$$

   for all $v_1, v_2 \in \mathcal{V}$. With $E(v, p) = \Phi(v, p) - \Omega(p)$, we obtain

$$\gamma\|p^\star(v_2) - p^\star(v_1)\|^2 \leq \langle p^\star(v_2) - p^\star(v_1), \nabla_2 \Phi(v_2, p^\star(v_2)) - \nabla_2 \Phi(v_1, p^\star(v_2)) \rangle.$$

   Since $\Phi$ is $\beta$-smooth, we have for all $p \in \mathcal{C}$

$$\|\nabla_2 \Phi(v_2, p) - \nabla_2 \Phi(v_1, p)\|_* \leq \beta\|(v_2, p) - (v_1, p)\| = \beta\|v_2 - v_1\|.$$

Combined with the Holder inequality $\langle a, b \rangle \leq \|a\|_* \|b\|$, this gives

$$\gamma \|p^\star(v_2) - p^\star(v_1)\|^2 \leq \beta \|p^\star(v_2) - p^\star(v_1)\| \|v_2 - v_1\|.$$

Simplifying, we get

$$\|p^\star(v_2) - p^\star(v_1)\| \leq \frac{\beta}{\gamma} \|v_2 - v_1\|. \tag{21}$$

Therefore, $p_\Omega^\Phi(v) = p^\star(v)$ is $\beta/\gamma$-Lipschitz.

From Rockafellar's envelope theorem [62, Theorem 10.31], $\nabla \Omega^\Phi(v) = \nabla_1 \Phi(v, p^\star(v))$. Since $\Phi$ is $\beta$-smooth, we therefore have

$$
\begin{aligned}
\|\nabla \Omega^\Phi(v_2) - \nabla \Omega^\Phi(v_1)\|_* &= \|\nabla_1 \Phi(v_2, p^\star(v_2)) - \nabla_1 \Phi(v_1, p^\star(v_1))\|_* \\
&\leq \beta \|(v_2 - p^\star(v_2)) - (v_1, p^\star(v_1))\| \\
&\leq \beta(\|v_2 - v_1\| + \|p^\star(v_2) - p^\star(v_1)\|) \\
&\leq \left( \beta + \frac{\beta^2}{\gamma} \right) \|v_2 - v_1\| \\
&\leq 2 \frac{\beta^2}{\gamma} \|v_2 - v_1\|,
\end{aligned}
$$

where we used (21) and $\frac{\beta}{\gamma} \geq 1$.

A related result in the context of bilevel programming but with different assumptions and proof is stated in [31, Lemma 2.2]. More precisely, the proof of that result requires twice differentiability of $E(v, p) = \Phi(v, p) - \Omega(p)$ while we do not. Moreover, applying that result to our setting would require $E(v, p)$ to be smooth in $(v, p)$ while we only assume $\Phi(v, p)$ to be the case. We emphasize again that $\Omega(p)$ is nonsmooth when it is the negentropy.

### B.2 Proofs for Proposition 2 (closed forms)

1. **Bilinear coupling.** This follows from

$$\Omega^\Phi(v) = \max_{p \in \mathcal{C}} \langle v, Up \rangle - \Omega(p) = \max_{p \in \mathcal{C}} \langle U^\top v, p \rangle - \Omega(p) = \Omega^*(U^\top v)$$

and similarly for $p_\Omega(v)$.

2. **Linear-quadratic coupling.** Let us define the function

$$F(p) = \frac{\gamma}{2} \|p\|_2^2 - \frac{1}{2} \langle p, Ap \rangle - \langle p, b \rangle = \frac{1}{2} \langle p, (\gamma I - A)p \rangle - \langle p, b \rangle.$$

Its gradient is $\nabla F(p) = (\gamma I - A)p - b$. Setting $\nabla F(p^\star) = 0$, we obtain

$$(\gamma I - A)p^\star = b \Leftrightarrow p^\star = (\gamma I - A)^{-1} b,$$

where we assumed that $(\gamma I - A)$ is positive definite. We therefore get

$$F(p^\star) = \frac{1}{2} \langle p^\star, b \rangle - \langle p^\star, b \rangle = -\frac{1}{2} \langle p^\star, b \rangle = -\frac{1}{2} \langle b, (\gamma I - A)^{-1} b \rangle.$$

3. **Metric coupling.** From (8), it is easy to check that $\Omega = -\Lambda \Leftrightarrow \Omega^\Phi = -\Lambda_C$ with $C = -\Phi$. From [60, Proposition 6.1], we have $\Lambda_C = -\Lambda$. Therefore, $\Omega^\Phi = -\Omega$.

### B.3 Proofs for Proposition 3 (Properties of generalized Fenchel-Young losses)

1. **Non-negativity.** This follows immediately from the generalized Fenchel-Young inequality.

2. **Zero loss.** If $p_\Omega^\Phi(v) = y$, then using $L_\Omega^\Phi(v, y) = F(v, y) - F(v, p_\Omega^\Phi(v))$, where $F(v, p) = \Omega(p) - \Phi(v, p)$, we obtain $L_\Omega^\Phi(v, y) = 0$. Let's now prove the reverse direction. Since the maximum in (5) exists, we have $F(v, p_\Omega^\Phi(v)) \leq F(v, y)$. If $L_\Omega^\Phi(v, y) = 0$, we have $F(v, p_\Omega^\Phi(v)) = F(v, y)$. Since the maximum is unique by assumption, this proves that $y = p_\Omega^\Phi(v)$.

3. **Gradient.** This follows directly from the definition (9).

4. **Difference of convex (DC).** If $\Phi(v, p)$ is convex in $p$, $\Omega^{\Phi}(v)$ is convex (Proposition 1). Therefore, $v \mapsto \Omega^{\Phi}(v) - \Phi(v, p)$ for all $p \in \mathcal{C}$.

5. **Smaller output set, smaller loss.** Let $\Omega\colon \mathcal{C} \to \mathbb{R}$ and let $\Omega'$ be the restriction of $\Omega$ to $\mathcal{C}' \subseteq \mathcal{C}$, i.e., $\Omega'(p) := \Omega + I_{\mathcal{C}'}$, where $I_{\mathcal{C}'}$ is the indicator function of $\mathcal{C}'$. From (5), we have $(\Omega')^{\Phi}(v) \leq \Omega^{\Phi}(v)$ for all $v \in \mathcal{V}$. From (9), we therefore have $L_{\Omega'}^{\Phi}(v, p) \leq L_{\Omega}^{\Phi}(v, p)$ for all $v \in \mathcal{V}$ and all $p \in \mathcal{C}'$.

6. **Quadratic lower-bound.** If a function $F$ is $\gamma$-strongly convex over $\mathcal{C}$ w.r.t. a norm $\|\cdot\|$, then

$$\frac{\gamma}{2}\|p - p'\|^2 \leq F(p) - F(p') - \langle \nabla F(p'), p - p' \rangle \tag{22}$$

for all $p, p' \in \mathcal{C}$. If $\mathcal{C}$ is a closed convex set, we also have that $p^\star = \operatorname{argmin}_{p \in \mathcal{C}} F(p)$ satisfies the optimality condition

$$\langle \nabla F(p^\star), p - p^\star \rangle \geq 0 \tag{23}$$

for all $p \in \mathcal{C}$ [53, Eq. (2.2.13)]. Combining (22) with $p' = p^\star$ and (23), we obtain

$$\frac{\gamma}{2}\|p - p^\star\|^2 \leq F(p) - F(p^\star).$$

Applying the above with $F(p) = \Omega(p) - \Phi(v, p)$ and using $p^\star = p_{\Omega}^{\Phi}(v)$, we obtain

$$\frac{\gamma}{2}\|y - p_{\Omega}^{\Phi}(v)\|^2 \leq L_{\Omega}^{\Phi}(v, y).$$

7. **Upper-bounds.** If $F$ is $\alpha$-Lipschitz over $\mathcal{C}$ with respect to a norm $\|\cdot\|$, then for all $p, p' \in \mathcal{C}$

$$|F(p) - F(p')| \leq \alpha\|p - p'\|.$$

Applying the above with $F(p) = \Omega(p) - \Phi(v, p)$ and $p' = p_{\Omega}^{\Phi}(v)$, we obtain

$$L_{\Omega}^{\Phi}(v, p) \leq \alpha\|p - p_{\Omega}^{\Phi}(v)\|.$$

If $\Phi(v, p)$ is concave in $p$, then its linear approximation always lies above:

$$\Phi(v, p') \leq \Phi(v, p) + \langle \nabla_2 \Phi(v, p), p' - p \rangle,$$

for all $p, p' \in \mathcal{C}$. We then have

$$\begin{aligned}
L_{\Omega}^{\Phi}(v, p) &= \Omega^{\Phi}(v) + \Omega(p) - \Phi(v, p) \\
&= \max_{p' \in \mathcal{C}} \ \Phi(v, p') - \Omega(p') + \Omega(p) - \Phi(v, p) \\
&\leq \max_{p' \in \mathcal{C}} \ \langle \nabla_2 \Phi(v, p), p' \rangle - \Omega(p') + \Omega(p) - \langle \nabla_2 \Phi(v, p), p \rangle \\
&= \Omega^*(\nabla_2 \Phi(v, p)) + \Omega(p) - \langle \nabla_2 \Phi(v, p), p \rangle \\
&= L_{\Omega}(\nabla_2 \Phi(v, p), p).
\end{aligned}$$

### B.4 Proof of Proposition 4 (calibration)

**Background.** The pointwise target risk of $\widehat{y} \in \mathcal{Y}$ according to $q \in \triangle^{|\mathcal{Y}|}$ is

$$\ell(\widehat{y}, q) := \mathbb{E}_{Y \sim q} L(\widehat{y}, Y).$$

We also define the corresponding excess of pointwise risk, the difference between the pointwise risk and the pointwise Bayes risk:

$$\delta\ell(\widehat{y}, q) := \ell(\widehat{y}, q) - \min_{y' \in \mathcal{Y}} \ell(y', q).$$

We can then write the risk of $f\colon \mathcal{X} \to \mathcal{Y}$ in terms of the pointwise risk

$$\begin{aligned}
\mathcal{L}(f) &:= \mathbb{E}_{(X, Y) \sim \rho} L(f(X), Y) \\
&= \mathbb{E}_{X \sim \rho_{\mathcal{X}}} \mathbb{E}_{Y \sim \rho(\cdot|X)} L(f(X), Y) \\
&= \mathbb{E}_{X \sim \rho_{\mathcal{X}}} \ell(f(X), \rho(\cdot|X)).
\end{aligned}$$

Let us define the Bayes predictor

$$f^\star := \operatorname*{argmin}_{f\colon \mathcal{X} \to \mathcal{Y}} \mathcal{L}(f).$$

The Bayes risk is then

$$\mathcal{L}(f^\star) = \min_{f\colon \mathcal{X}\to\mathcal{Y}} \mathbb{E}_{X\sim\rho_{\mathcal{X}}}\mathbb{E}_{Y\sim\rho(\cdot|X)}L(f(X),Y)$$

$$= \mathbb{E}_{X\sim\rho_{\mathcal{X}}}\min_{y'\in\mathcal{Y}}\mathbb{E}_{Y\sim\rho(\cdot|X)}L(y',Y)$$

$$= \mathbb{E}_{X\sim\rho_{\mathcal{X}}}\min_{y'\in\mathcal{Y}}\ell(y',\rho(\cdot|X)).$$

Combining the above, we can write the excess of risk of $f\colon \mathcal{X}\to\mathcal{Y}$ as

$$\mathcal{L}(f) - \mathcal{L}(f^\star) = \mathbb{E}_{X\sim\rho_{\mathcal{X}}}\delta\ell(f(X),\rho(\cdot|X)).$$

Similarly, with the generalized Fenchel-Young loss (9), the pointwise surrogate risk of $v\in\mathcal{V}$ according to $q\in\triangle^{|\mathcal{Y}|}$ is

$$\ell_\Omega^\Phi(v,q) := \mathbb{E}_{Y\sim q}\,L_\Omega^\Phi(v,Y)$$

and the excess of pointwise surrogate risk is

$$\delta\ell_\Omega^\Phi(v,q) := \ell_\Omega^\Phi(v,q) - \min_{v'\in\mathcal{V}}\ell_\Omega^\Phi(v',q).$$

Let us define the surrogate risk of $g\colon \mathcal{X}\to\mathcal{V}$ by

$$\mathcal{L}_\Omega^\Phi(g) := \mathbb{E}_{(X,Y)\sim\rho}\,L_\Omega^\Phi(g(X),Y)$$

and the corresponding Bayes predictor by

$$g^\star := \operatorname*{argmin}_{g\colon \mathcal{X}\to\mathcal{V}}\mathcal{L}_\Omega^\Phi(g).$$

We can then write the excess of surrogate risk of $g\colon \mathcal{X}\to\mathcal{V}$ as

$$\mathcal{L}_\Omega^\Phi(g) - \mathcal{L}_\Omega^\Phi(g^\star) = \mathbb{E}_{X\sim\rho_{\mathcal{X}}}\delta\ell_\Omega^\Phi(g(X),\rho(\cdot|X)).$$

A calibration function [68] $\xi\colon \mathbb{R}_+ \to \mathbb{R}_+$ is a function relating the excess of pointwise target risk and pointwise surrogate risk. It should be non-negative, convex and non-decreasing on $\mathbb{R}_+$, and satisfy $\xi(0)=0$. Formally, given a decoder $d\colon \mathcal{V}\to\mathcal{Y}$, $\xi$ should satisfy

$$\xi(\delta\ell(d(v),q)) \le \delta\ell_\Omega^\Phi(v,q) \tag{24}$$

for all $v\in\mathcal{V}$ and $q\in\triangle^{|\mathcal{Y}|}$. By Jensen's inequality, this implies that the target and surrogate risks are calibrated for all $g\colon \mathcal{X}\to\mathcal{V}$ [59, 56]

$$\xi(\mathcal{L}(d\circ g) - \mathcal{L}(f^\star)) \le \mathcal{L}_\Omega^\Phi(g) - \mathcal{L}_\Omega^\Phi(g^\star).$$

From now on, we can therefore focus on proving (24).

**Upper-bound on the pointwise target excess risk.**    We now make use of the affine decomposition (12). Let $\sigma := \sup_{y\in\mathcal{Y}}\|V^\top\varphi(y)\|_*$, where $\|\cdot\|_*$ denotes the dual norm of $\|\cdot\|$. Let us define

$$\tilde{y}_L(u) := \operatorname*{argmin}_{\widehat{y}\in\mathcal{Y}}\langle\varphi(\widehat{y}),Vu+b\rangle$$

and $\mu_\varphi(q) := \mathbb{E}_{Y\sim q}[\varphi(Y)] \in \operatorname{conv}(\varphi(\mathcal{Y})) \subseteq \varphi(\mathbb{R}^k)$. From [16, Lemma 2],

$$\delta\ell(\tilde{y}_L(u),q) \le 2\sigma\|\mu_\varphi(q)-u\| \quad \forall u\in\varphi(\mathbb{R}^k), q\in\triangle^{|\mathcal{Y}|}.$$

Using $y_L(p) = \tilde{y}_L(u)$ with $u=\varphi(p)$, we thus get

$$\delta\ell(y_L(p),q) \le 2\sigma\|\mu_\varphi(q)-\varphi(p)\| \quad \forall p\in\mathbb{R}^k, q\in\triangle^{|\mathcal{Y}|}. \tag{25}$$

**Bound on the pointwise surrogate excess risk (bilinear case).** To highlight the difference with our proof technique, we first prove the result in the bilinear case, assuming $\Omega$ is $\gamma$-strongly convex. We follow the same proof technique as [56, 16] but unlike these works we do not require any Legendre-type assumption on $\Omega$. If $\Phi(v, p) = \langle v, \varphi(p) \rangle = \langle v, p \rangle$, we have

$$
\begin{aligned}
\ell_\Omega^\Phi(v, q) &= \mathbb{E}_{Y \sim q}\, L_\Omega(v, Y) \\
&= \Omega^*(v) + \mathbb{E}_{Y \sim q} \Omega(Y) - \langle v, \mu(q) \rangle \\
&= \Omega^*(v) + \Omega(\mu(q)) - \langle v, \mu(q) \rangle + \mathbb{E}_{Y \sim q} \Omega(Y) - \Omega(\mu(q)) \\
&= L_\Omega(v, \mu(q)) + \mathbb{E}_{Y \sim q} \Omega(Y) - \Omega(\mu(q)),
\end{aligned}
$$

where, when $\varphi(y) = y$, we denote $\mu(q) := \mathbb{E}_{Y \sim q}[Y] \in \operatorname{conv}(\mathcal{Y}) \subseteq \mathbb{R}^k$ for short. The quantity $\mathbb{E}_{Y \sim q} \Omega(Y) - \Omega(\mu(q))$ is called Bregman information [7] or Jensen gap, and is non-negative if $\Omega$ is convex. This term cancels out in the excess of pointwise surrogate risk

$$
\delta\ell_\Omega(v, q) := \ell_\Omega(v, q) - \min_{v' \in \mathcal{V}} \ell_\Omega(v', q) = L_\Omega(v, \mu(q)) - \min_{v' \in \mathcal{V}} L_\Omega(v', \mu(q)).
$$

Since the Fenchel-Young loss achieves its minimum at 0, we have

$$
\delta\ell_\Omega(v, q) = L_\Omega(v, \mu(q)).
$$

Therefore, the excess of pointwise surrogate risk can be written in Fenchel-Young loss form. By the quadratic lower-bound (10) and the upper-bound (25), we have

$$
\delta\ell_\Omega(v, q) = L_\Omega(v, \mu(q)) \geq \frac{\gamma}{2} \|\mu(q) - p_\Omega(v)\|^2 \geq \frac{\gamma}{8\sigma^2} \delta\ell(y_L(p_\Omega(v)), q))^2.
$$

Therefore the calibration function with the decoder $d = y_L \circ p_\Omega$ is

$$
\xi(\varepsilon) = \frac{\gamma \varepsilon^2}{8\sigma^2}.
$$

**Bound on the pointwise surrogate excess risk (linear-concave case).** We now prove the bound assuming that $L_\Omega^\Phi$ is smooth and $\Phi(v, p) = \langle v, \varphi(p) \rangle$. This includes the previous proof as special case because when $\Phi(v, p) = \langle v, p \rangle$, then $L_\Omega^\Phi(v, y) = L_\Omega(v, y)$ is $\frac{1}{\gamma}$-smooth in $v$ if and only if $\Omega(p)$ is $\gamma$-strongly convex in $p$ (cf. §2).

By Theorem 4.22 in [57], if a function $f(v)$ is $M$-smooth in $v$ w.r.t. the dual norm $\| \cdot \|_*$ and is bounded below, then

$$
f(v) - \min_{v' \in \mathcal{V}} f(v') \geq \frac{1}{2M} \|\nabla f(v)\|^2.
$$

With $f(v) = \ell_\Omega^\Phi(v, q) = \mathbb{E}_{Y \sim q}\, L_\Omega(v, Y)$, which is non-negative, we obtain

$$
\begin{aligned}
\delta\ell_\Omega^\Phi(v, q) &:= \ell_\Omega^\Phi(v, q) - \min_{v' \in \mathcal{V}} \ell_\Omega^\Phi(v', q) \\
&\geq \frac{1}{2M} \|\nabla_1 \ell_\Omega^\Phi(v, q)\|^2 \\
&= \frac{1}{2M} \|\mathbb{E}_{Y \sim q} \nabla_1 L_\Omega^\Phi(v, Y)\|^2 \\
&= \frac{1}{2M} \|\nabla \Omega^\Phi(v) - \mathbb{E}_{Y \sim q} \nabla_1 \Phi(v, Y)\|^2 \\
&= \frac{1}{2M} \|\nabla_1 \Phi(v, p_\Omega^\Phi(v)) - \mathbb{E}_{Y \sim q} \nabla_1 \Phi(v, Y)\|^2 \\
&= \frac{1}{2M} \|\varphi(p_\Omega^\Phi(v)) - \mathbb{E}_{Y \sim q} \varphi(Y)\|^2 \\
&= \frac{1}{2M} \|\varphi(p_\Omega^\Phi(v)) - \mu_\varphi(q)\|^2.
\end{aligned}
$$

We therefore have

$$
\delta\ell_\Omega^\Phi(v, q) \geq \frac{1}{2M} \|\mu_\varphi(q) - \varphi(p_\Omega^\Phi(v))\|^2 \geq \frac{1}{8\sigma^2 M} \delta\ell(y_L(p_\Omega^\Phi(v)), q))^2.
$$

Therefore the calibration function with the decoder $d = y_L \circ p_\Omega^\Phi$ is

$$
\xi(\varepsilon) = \frac{\varepsilon^2}{8\sigma^2 M}.
$$

### B.5 Proofs for Proposition 5 (Properties of generalized biconjugates)

The proofs are similar to that for $C$-transforms [65, Proposition 1.34].

1. **Lower bound.** Let $\Omega \colon \mathcal{C} \to \mathbb{R}$. We have

$$
\begin{aligned}
\Omega^{\Phi\Psi}(p) &\coloneqq \max_{v \in \mathcal{V}} \Psi(p, v) - \Omega^{\Phi}(v) \\
&= \max_{v \in \mathcal{V}} \Phi(v, p) - \Omega^{\Phi}(v) \\
&= \max_{v \in \mathcal{V}} \Phi(v, p) - \left[ \max_{p' \in \mathcal{C}} \Phi(v, p') - \Omega(p') \right] \\
&\leq \max_{v \in \mathcal{V}} \Phi(v, p) - \Phi(v, p) + \Omega(p) \\
&= \Omega(p).
\end{aligned}
$$

   Therefore, $\Omega^{\Phi\Psi}(p) \leq \Omega(p)$ for all $p \in \mathcal{C}$. Analogously, for a function $\Lambda \colon \mathcal{V} \to \mathbb{R}$, we have $\Lambda^{\Psi\Phi}(v) \leq \Lambda(v)$ for all $v \in \mathcal{V}$.

2. **Equality.** Since $\Omega$ is $\Psi$-convex, there exits $\Lambda \colon \mathcal{V} \to \mathbb{R}$ such that $\Omega = \Lambda^{\Psi}$. We then have $\Omega^{\Phi} = \Lambda^{\Psi\Phi}$. Using the lower bound property, we get $\Omega^{\Phi}(v) = \Lambda^{\Psi\Phi}(v) \leq \Lambda(v)$ for all $v \in \mathcal{V}$. By the order reversing property, we have $\Omega^{\Phi\Psi}(p) \geq \Lambda^{\Psi}(p) = \Omega(p)$ for all $p \in \mathcal{C}$. However, we also have $\Omega^{\Phi\Psi}(p) \leq \Omega(p)$ for all $p \in \mathcal{C}$. Therefore, $\Omega^{\Phi\Psi}(p) = \Omega(p)$ for all $p \in \mathcal{C}$.

3. **Tightest lower-bound.** Let $\Omega'$ be any lower bound of $\Omega$ that is $\Psi$-convex. Therefore, there exists $\Lambda \colon \mathcal{V} \to \mathbb{R}$ such that $\Omega'(p) = \Lambda^{\Psi}(p) \leq \Omega(p)$ for all $p \in \mathcal{C}$. By the order reversing property, we have $\Lambda^{\Psi\Phi}(v) \geq \Omega^{\Phi}(v)$ for all $v \in \mathcal{V}$. By the lower bound property, we also have $\Lambda^{\Psi\Phi}(v) \leq \Lambda(v)$ for all $v \in \mathcal{V}$ and therefore, $\Lambda(v) \geq \Omega^{\Phi}(v)$ for all $v \in \mathcal{V}$. Applying the order reversing property once more, we get $\Omega'(p) = \Lambda^{\Psi}(p) \leq \Omega^{\Phi\Psi}(p)$ for all $p \in \mathcal{C}$. Therefore $\Omega^{\Phi\Psi}$ is the tightest lower bound of $\Omega$.

### B.6 Proofs for Proposition 6 (Properties of generalized Bregman divergences)

1. **Link with generalized Fenchel-Young losses.** From (16) and (17), we have

$$
\Lambda^{\Psi}(p) = \Phi(v_{\Lambda}^{\Psi}(p), p) - \Lambda(v_{\Lambda}^{\Psi}(p)) \quad \forall p \in \mathcal{C}
$$

   for any $\Lambda \colon \mathcal{V} \to \mathbb{R}$. With $\Lambda = \Omega^{\Phi}$, if $\Omega$ is $\Phi$-convex, using Proposition 5, we have

$$
\Omega^{\Phi\Psi}(p) = \Omega(p) = \Phi(v_{\Lambda}^{\Psi}(p), p) - \Lambda(v_{\Lambda}^{\Psi}(p)) \quad \forall p \in \mathcal{C}.
$$

   Plugging $\Omega(p')$ in (18) and using the shorthand $v \coloneqq v_{\Omega^{\Phi}}^{\Psi}(p')$, we get

$$
D_{\Omega}^{\Phi}(p, p') = \Omega(y) - \Phi(v, p) + \Omega^{\Phi}(v) = L_{\Omega}^{\Phi}(v, p).
$$

2. **Non-negativity.** This follows directly from the non-negativity of $L_{\Omega}^{\Phi}(v, p)$.

3. **Identity of indiscernibles**. If $p = p'$, we immediately obtain $D_{\Omega}^{\Phi}(p, p') = 0$ from (18). Let's prove the reverse direction. If $D_{\Omega}^{\Phi}(p, p') = 0$, we have $F(v_{\Omega^{\Phi}}^{\Psi}(p'), p) = F(v_{\Omega^{\Phi}}^{\Psi}(p'), p')$ where $F(v, p) = \Omega(p) - \Phi(v, p)$. Since by assumption $F$ is strictly convex in $p$, we obtain $p = p'$.

4. **Convexity.** This follows from $D_{\Omega}^{\Phi}(p, p') = \Omega(p) - \Phi(v, p) + \text{const}$.

5. **Recovering Bregman divergences.** If $\Phi(v, p) = \langle v, p \rangle$, we have

$$
v_{\Omega^{\Phi}}^{\Psi}(p) = \operatorname*{argmax}_{v \in \mathcal{V}} \Phi(v, p) - \Omega^{\Phi}(v) = \operatorname*{argmax}_{v \in \mathcal{V}} \langle v, p \rangle - \Omega^{*}(v) = \nabla\Omega(p).
$$

   Plugging back in (18), we obtain the Bregman divergence (20).

# C Experimental details

## C.1 Multilabel classification

**Datasets.** We used public datasets available at `https://www.csie.ntu.edu.tw/~cjlin/libsvmtools/datasets/`. Dataset statistics are summarized in Table 3. For all datasets, we normalize samples to have zero mean unit variance.

Table 3: Multilabel dataset statistics.

| Dataset | Type | Train | Dev | Test | Features | Classes | Avg. labels |
|---|---|---|---|---|---|---|---|
| Birds | Audio | 134 | 45 | 172 | 260 | 19 | 1.96 |
| Cal500 | Music | 376 | 126 | 101 | 68 | 174 | 25.98 |
| Emotions | Music | 293 | 98 | 202 | 72 | 6 | 1.82 |
| Mediamill | Video | 22,353 | 7,451 | 12,373 | 120 | 101 | 4.54 |
| Scene | Images | 908 | 303 | 1,196 | 294 | 6 | 1.06 |
| Yeast | Micro-array | 1,125 | 375 | 917 | 103 | 14 | 4.17 |

**Experimental details.** In all experiments, we set the activation $\sigma$ to $\mathrm{relu}(a) \coloneqq \max\{0, a\}$.

For the unary model, we use a neural network with one hidden-layer, i.e., $g_\theta(x) = W_2\sigma(W_1 x + b_1) + b_2$, where $\theta = (W_2, b_2, W_1, b_1)$, $W_2 \in \mathbb{R}^{k \times m}$, $b_2 \in \mathbb{R}^k$, $W_1 \in \mathbb{R}^{m \times d}$, $b_1 \in \mathbb{R}^m$, and $m$ is the number of hidden units. We use the heuristic $m = \min\{100, d/3\}$, where $d$ is the dimensionality of $x$.

For the pairwise model, in order to obtain a negative semi-definite matrix $U$, we parametrize $U = -AA^\top$ with $A = [W_1 x + b_1, \ldots, W_m x + b_m]$, where $W_j \in \mathbb{R}^{k \times d}$ and $b_j \in \mathbb{R}^k$ In our experiments, we choose a rank-one model, i.e., $m = 1$. Note that we use distinct parameters for the unary and pairwise models but sharing parameters would be possible.

For the SPEN model, following [11, Eq. 4 and 5], we set the energy to $\Phi(v, p) = \langle u, p \rangle - \Psi(w, p)$, where $v = (u, w)$, $u = g_\theta(x)$ and $w$ are the weights of the "prior network" $\Psi$ (independent of $x$). We parametrize $\Psi(w, p) = W_2\sigma(W_1 p + b_1) + b_2$, where $w = (W_2, b_2, W_1, b_1)$, $W_2 \in \mathbb{R}^{1 \times m}$, $b_2 \in \mathbb{R}^1$ and $W_1 \in \mathbb{R}^{m \times k}$, $b_1 \in \mathbb{R}^m$. To further impose convexity of $\Psi$ in $p$, $W_2$ needs to be non-negative. To do so without using constrained optimization, we use the change of variable $W_2 = \mathrm{softplus}(W_2')$, where $\mathrm{softplus}(a) \coloneqq \log(1 + \exp(a))$ is used as an element-wise bijective mapping.

**Additional results.** The gradient of $\Omega^\Phi(v)$ can be computed using the envelope theorem (Proposition 1), which does not require to differentiate through $p_\Omega^\Phi(v)$. Alternatively, since $\Omega^\Phi(v) = \Phi(v, p_\Omega^\Phi(v)) - \Omega(p_\Omega^\Phi(v))$, we can also compute the gradient of $\Omega^\Phi(v)$ by using the implicit function theorem, differentiating through $p_\Omega^\Phi(v)$. To do so, we use the approach detailed in [17]. Results in Table 4 show that the envelope theorem performs comparably to the implicit function theorem, if not slightly better. Differentiating through $p_\Omega^\Phi(v)$ using the implicit function theorem requires to solve a $k \times k$ system and its implementation is more complicated than the envelope theorem. Therefore, we suggest to use the envelope theorem in practice.

Table 4: Comparison of envelope and implicit function theorems on the pairwise model (test accuracy in %).

| | yeast | scene | mediamill | birds | emotions | cal500 |
|---|---|---|---|---|---|---|
| Envelope theorem | 80.19 | **91.58** | **96.95** | **91.55** | **80.56** | **85.73** |
| Implicit function theorem | **80.33** | **91.58** | **96.95** | 91.54 | 80.53 | 85.57 |

In addition, we also compared the proposed generalized Fenchel-Young loss with the energy loss, the binary cross-entropy loss and the generalized perceptron loss, which corresponds to setting $\Omega(p) = 0$. As explained in §3, the cross-entropy loss requires to differentiate through $p_\Omega^\Phi(v)$; we do so by implicit differentiation. Table 5 shows that the generalized Fenchel-Young loss outperforms these losses. As expected, the energy loss performs very poorly, as it can only push the model in one direction [44]. Using regularization $\Omega$, as advocated in this paper, is empirically confirmed to be beneficial for accuracy.

Table 5: Comparison of loss functions for the pairwise model (accuracy in %).

|  | yeast | scene | mediamill | birds | emotions | cal500 |
|---|---|---|---|---|---|---|
| Generalized FY loss | **80.19** | **91.58** | **96.95** | 91.55 | **80.56** | 85.73 |
| Energy loss | 42.35 | 33.02 | 40.92 | 14.29 | 55.50 | 39.27 |
| Cross-entropy loss | 79.00 | 90.78 | 96.77 | **91.56** | 78.08 | **85.89** |
| Generalized perceptron loss | 68.36 | 89.33 | 93.24 | 88.92 | 66.34 | 80.11 |

## C.2 Imitation learning

**Experimental details.** We run a hyperparameter search over the learning rate of the ADAM optimizer, the number of hidden units in the layers, the weight of the L2 parameters regularization term and the scale of the energy regularization term $\Omega$. We run the hyperparameter search for 4 demonstration trajectories and select the best performing ones based on the final performance (averaged over 3 seeds).

Table 6: Hyperparameter search for imitation learning.

| Model | Learning rate {1e-4, 5e-4, 1e-3} | Params regularization {0., 1., 10.} | Energy regularization {0.1, 1., 10.} | Hidden units {16, 32, 64, 128} |
|---|---|---|---|---|
| Unary | 5e-4 | 0.0 | 1. | 16 |
| Pairwise | 1e-4 | 0.0 | 10. | 32 |

**Environments.** We also provide performance of the expert agent as detailed by Orsini et al. [58] as well as the description of the observation and action spaces for each environment.

Table 7: Dimension of observation space, dimension of action space, expert performance, and random policy performance for each environment.

| Task | Observations | Actions | Random policy score | Expert score |
|---|---|---|---|---|
| HalfCheetah-v2 | 17 | 6 | -282 | 8770 |
| Hopper-v2 | 11 | 3 | 18 | 2798 |
| Walker-v2 | 17 | 6 | 1.6 | 4118 |
| Ant-v2 | 111 | 8 | 123 | 5637 |