# OpenReview forum: "Learning Energy Networks with Generalized Fenchel-Young Losses"
_NeurIPS.cc/2022/Conference — NeurIPS 2022 Accept_

### Official Review · Reviewer_8vqD · 2022-07-11

**Rating:** 7
**Confidence:** 3
**Soundness:** 4 excellent
**Presentation:** 4 excellent
**Contribution:** 4 excellent

**Summary:**

The most common approaches to training EBMs are approximate approaches, where the approximation comes from using MCMC to sample from the EBM. This is because training an EBM using the exact gradients is intractable, due to differentiation through argmax/argmin. This paper constructs new loss functions that can be used to tractably train an EBM using exact gradients.

The authors apply their approach to imitation learning and multilabel classification.

Note: I am not very familiar with theory, so I did not read the math sections closely.


**Questions:**

None at this time

**Limitations:**

I do not see any potential negative societal impact of this work.

**Strengths And Weaknesses:**

The experiments are done with great thoroughness. The details for reproducibility are included, and the experiments are done in a robust manner (e.g. averaging over multiple seeds and properly tuning hyperparameters on validation sets).

The authors show that they are able to achieve competitive performance on multilabel classification, despite using energy networks that aren’t arbitrarily nonlinear. In other words, they show that the losses arising from the energy networks they use in these experiments are expressive enough to be accurate.

---

> ### Author Response · Authors · 2022-07-28
> **Thank you for your review**
>
> > The most common approaches to training EBMs are approximate approaches, where the approximation comes from using MCMC to sample from the EBM. This is because training an EBM using the exact gradients is intractable, due to differentiation through argmax/argmin. This paper constructs new loss functions that can be used to tractably train an EBM using exact gradients.
>
> This is a good summary of our paper and indeed we do not require MCMC techniques, unlike existing approaches for EBMs.
>
> > The experiments are done with great thoroughness. The details for reproducibility are included, and the experiments are done in a robust manner (e.g. averaging over multiple seeds and properly tuning hyperparameters on validation sets).
>
> Thank you very much for the very positive review.

---

### Official Review · Reviewer_jRDG · 2022-07-11

**Rating:** 5
**Confidence:** 4
**Soundness:** 3 good
**Presentation:** 3 good
**Contribution:** 2 fair

**Summary:**

The paper introduces a class of losses based on a generalized Fenchel-Young inequality for learning energy models. The idea is to maximize the agreement between  the target $y$ and a feature $v = f(x)$ obtained using a model $f$ applied to an input $x$. The agreement is measured by minimizing an energy function $\phi(v,y)$ to which a regularization $\Omega(y)$ is added. Hence, for a given a feature $v$, the corresponding prediction $p(v)$ is given as the maximizer of the objective $\max_p F(v,p) := \phi(v,p)-\Omega(p)$. The model $f$ learned by minimizing the discrepancy between   $F(v ,y)$ and the optimal value $F(v,p(v))$. In other words, minimizing $L(v,y) = F(v,p(v))-F(v ,y)$.

The paper shows, amongst other properties, that such objectives $L$ can be optimized without the need to differentiate wrt the optimal $p(v)$ and can thus be implemented easily.

Simple experiments show that the approach allows to consider more general losses that can yield improvement on classification and imitation learning tasks.


**Questions:**

- The paragraph in L140-148 is a bit confusing: what do the authors mean by measuring the discrepancy between p_{\Omega} and $y$ without regularization Omega? Isn’t p_{\Omega} defined by a choice of a function Omega?


- Envelope thm: The authors refer to some assumptions in the appendix under which the envelope theorem holds. These assumptions are distilled in the text of the proof. They should be explicitly stated somewhere (ideal in the main text).









**Limitations:**

The proposed approach has a rather limited novelty (both technical and conceptual). However, this can be mitigated if the authors provide more evidence for the significance of such generalization in practice.

**Strengths And Weaknesses:**

Strenghts:
The paper is clearly written and the results and derivations are sound. The experiments, although basic, show a marginal improvement, especially in the context of imitation learning.

Weaknesses: Originality and Significance
Originality: The idea of extending Fenchel-Young loss, which uses an energy function given by a scalar product $ \phi(v,y)= <v,y>$ is rather direct and straightforward and does not represent a significant technical challenge. For instance, the proofs are often direct consequences of well-known results such as the envelop theorem in proposition 1.

Significance: Given that most of the experimental results show only a marginal improvement compared to more standard objectives, I am not fully convinced by the significance of the results. Do the authors have situations in mind where one requires such more general losses and where simpler approaches fail?

---

> ### Author Response · Authors · 2022-07-28
> **Summary of our mathematical contributions (not mentioned in the review)**
>
> Thank you for taking the time to review our paper. We believe your score mainly stems from the claim of "lacking novelty" and
> "being straightforward". We believe this is not justified, as we hope to convince you below.
>
> > The idea of extending Fenchel-Young loss, which uses an energy function given by a scalar product  is rather direct and straightforward and does not represent a significant technical challenge. For instance, the proofs are often direct consequences of well-known results such as the envelop theorem in proposition 1.
>
> It seems that this review ignores all of our key (mathematical) contributions:
> - We introduced the new notion of a regularized energy network.
> - Generalized conjugates are not well-known at all in the ML community. On the contrary, our paper will bring awareness to this new tool.
> - The smoothness result (Proposition 1, item 6) is new and not straightforward.
> - The lower bound result (Proposition 3, item 5) is more general than the existing one for regular FY losses and uses a simpler proof.
> - The calibration guarantees (Proposition 4) are more general than the existing ones for regular FY losses. They use a novel and not straightforward proof technique.
> - Generalized Bregman divergences (Appendix A) are completely new and not straightforward.
>
> The gradient computation indeed follows from envelope theorems (Danskin’s theorem if Phi is convex in v or Rockafellar’s theorem otherwise). We do not claim that this is difficult. Our main point is that our loss enjoys easy-to-compute gradients without argmax differentiation, thanks to this property.
>
> Overall, we believe that our paper advances the field of energy networks by introducing a principled loss construction with theoretical guarantees.
>
> > Given that most of the experimental results show only a marginal improvement compared to more standard objectives, I am not fully convinced by the significance of the results. Do the authors have situations in mind where one requires such more general losses and where simpler approaches fail?
>
> Our contribution is first and foremost a mathematical one. On the empirical side, while the improvements are indeed not big for the multilabel classification experiment, we argue they are significant in the imitation learning experiment. The pairwise model presented in the experiments is precisely a good example of an energy network enabled by our loss. Note that the goal of this paper is not to argue that energy networks are state-of-the-art for the tasks considered in our experiments but to introduce a better loss for learning them. This is confirmed empirically in Table 5, where we compared our loss with existing losses.
>
> > The paragraph in L140-148 is a bit confusing: what do the authors mean by measuring the discrepancy between p_{\Omega} and  without regularization Omega? Isn’t p_{\Omega} defined by a choice of a function Omega?
>
> Thank you, we agree this was confusing. By “without regularization”, we meant that Omega is 0. The regularization term Omega is independent of v, so it can be omitted in the energy loss. We added a clarification to the revised manuscript.
>
> > Envelope thm: The authors refer to some assumptions in the appendix under which the envelope theorem holds. These assumptions are distilled in the text of the proof. They should be explicitly stated somewhere (ideal in the main text).
>
> While we agree this should ideally be the case, this choice was made for space reasons because the assumptions are too long to state fully in the main text. Since we already mention that the precise assumptions are deferred to the appendix, we believe that this is an acceptable compromise.

---

> > ### Comment · Reviewer_jRDG · 2022-08-08
> > **Thank you for your response**
> >
> > I have read the author's response and the other reviews.
> >
> > Regarding the author's response to reviewer 8vqD: The authors claim that the proposed method does not require MCMC sampling unlike existing approaches to EBM. Often MCMC is used when using an EBM with a complex model which yields a non-convex objective and high-dimensional sampling problem. It is unclear to me what choice of \phi-functional would result in a ‘’more” tractable EBM objective that still results in an equally expressive model.  (Knowing that high dimensional sampling and non-convex optimization are both NP hard problems.)
> >
> >
> > - Avoiding argmax differentiation: It seems that one of the main arguments for the proposed losses is that they bypass the need for differentiating wrt the argmin. While this is a nice property, there are many prior works that proposed to rely on the envelope theorem, thus bypassing the need for differentiating through the argmax, especially in the context of generative models: [Bottou 2017, Geometrical insights for implicit generative modelling, Nowozing 2016 f-GANs]. Therefore, the discussions, for instance the one  in L140-150 should also be more nuanced about this.
> >
> >
> > - Statement of the assumption: I still couldn’t find where the assumptions are clearly stated (separately from the proof), they are still distilled inside the proof, which makes the reading difficult. The authors should state the assumptions outside of the proofs and refer to them.
> >
> >
> >
> > - Novelty:
> >     - Proposition 1 item 6: see lemma 2.2 (b) in Ghadimi 2018. This considers a gamma-strongly convex function g(v,p) in $p$ and that the hessian Nabla_{x,y} g(x,y) is bounded by beta. Can be applied to g(v,p)= Omega(p)-Phi(v,p) to directly obtain the  lipschitz smoothness of p(v) and then deduce that Omega(v)^{\phi} is (beta + beta^2/gamma) smooth.
> >
> >
> > - Overall, I am not convinced by the significance of the contribution and in particular what is gained by such a level of generalization.

---

> > > ### Author Response · Authors · 2022-08-08
> > > **Comments taken into account in revised manuscript**
> > >
> > > Thank you for engaging in the discussion. We have revised the paper to take into account your remarks.
> > >
> > > > Regarding the author's response to reviewer 8vqD: The authors claim that the proposed method does not require MCMC sampling unlike existing approaches to EBM. Often MCMC is used when using an EBM with a complex model which yields a non-convex objective and high-dimensional sampling problem. It is unclear to me what choice of \phi-functional would result in a ‘’more” tractable EBM objective that still results in an equally expressive model. (Knowing that high dimensional sampling and non-convex optimization are both NP hard problems.)
> > >
> > > The comment on high-dimensional sampling vs nonconvex optimization is a very valid one. We note however that MCMC techniques are typically used for probabilistic EBMs while we focus on EBMs in the original sense of LeCun (2006), i.e., networks with an argmax output layer. When the argmax objective is nonconcave in p, it can indeed be NP hard to solve the problem exactly and typically we can only get approximate gradients, as we clearly stated in the paper. The most general setting while still getting an optimization problem solvable in polynomial time (leading to exact or very accurate gradients) is when the energy is parametrized with an ICNN. We believe our proposed loss is a clear advance in this setting.
> > >
> > > > Avoiding argmax differentiation: It seems that one of the main arguments for the proposed losses is that they bypass the need for differentiating wrt the argmin. While this is a nice property, there are many prior works that proposed to rely on the envelope theorem, thus bypassing the need for differentiating through the argmax, especially in the context of generative models: [Bottou 2017, Geometrical insights for implicit generative modelling, Nowozing 2016 f-GANs]. Therefore, the discussions, for instance the one in L140-150 should also be more nuanced about this.
> > >
> > > We added the references in the revised manuscript for completeness but again we do not claim to be the first to use envelope theorems. We claim that our loss function is constructed in such a way that envelope theorems *can* be applied. When using arbitrary loss functions, envelope theorems can’t be applied and argmax differentiation is required instead. **If we follow the reviewer’s reasoning, any paper using envelope theorems is automatically not novel…**
> > >
> > > > Statement of the assumption: I still couldn’t find where the assumptions are clearly stated (separately from the proof), they are still distilled inside the proof, which makes the reading difficult. The authors should state the assumptions outside of the proofs and refer to them.
> > >
> > > We now added the assumptions directly in the main text, see revised manuscript. We also added more background information and the above suggested citation.
> > >
> > > > Proposition 1 item 6: see lemma 2.2 (b) in Ghadimi 2018. This considers a gamma-strongly convex function g(v,p) in  and that the hessian Nabla_{x,y} g(x,y) is bounded by beta. Can be applied to g(v,p)= Omega(p)-Phi(v,p) to directly obtain the lipschitz smoothness of p(v) and then deduce that Omega(v)^{\phi} is (beta + beta^2/gamma) smooth.
> > >
> > > This is an interesting reference that we added to the revised manuscript. However, the assumptions and the proof are different. Indeed, we do not require twice differentiability and we exploit the particular expression g(v,p)= Omega(p)-Phi(v,p). Indeed, applying the result of Ghadimi et al directly would require both Omega and Phi to be smooth in p, while we do not require the smoothness of Omega in p. This is important, as for example the negative entropy is not smooth. **In other words, we state a stronger result specialized for generalized conjugate functions.** We remind the reviewer that the smoothness property is essential to prove our key mathematical contribution, the calibration guarantees in Proposition 4.
> > >
> > > > Overall, I am not convinced by the significance of the contribution and in particular what is gained by such a level of generalization.
> > >
> > > This seems to be a criticism of energy networks in general rather than of our proposed loss function, i.e., does replacing the bilinear pairing with a more general energy function provably allow better generalization capability? We agree that studying this question is interesting but it’s clearly out-of-scope of this paper and our paper shouldn’t be rejected on such grounds.
> > >
> > > From a more philosophical point of view, we argue that one of the quests of mathematics is to aim for generality. It is enlightening to see that many properties of regular Fenchel-Young losses can be extended to a much more general setting.

---

> > > > ### Comment · Reviewer_jRDG · 2022-08-09
> > > > **Thank you, I still have some comments**
> > > >
> > > > Thank you for the clarifications, here are a few remarks:
> > > >   - **Argmax diff** "If we follow the reviewer’s reasoning, any paper using envelope theorems is automatically not novel". No, the point is to provide more context and to acknowledge prior works. As currently written, the reader has the impression that the present paper is the first to exploit the envelope thm in the context of ML, especially in the intro L27-42. The sentence the authors included: "For other envelope theorem usecases in machine learning, see, e.g., [21].", when discussing the assumptions does not really provide enough context for this. I think the references should be discussed earlier in the intro (L27-42). Please note that the losses arising in these references can be expressed an energy loss, so the connection to the present work is actually strong.
> > > >   - **Proposition 1 item 6:** It is good to mention that the authors mentioned the earlier result. I have a few comments though: the assumptions Ghadimi et al requires $\nabla_y\nabla_{x} g(x,y)$ to be bounded, this can arguably be easily relaxed to "$\nabla_{x} g(x,y)$ is $\beta$  Lipschitz" which does not need to make any smoothness assumption on $\Omega$. One can still say the assumptions are different, but please just be precise about how different they are (in L565).
> > > >   - "The most general setting while still getting an optimization problem solvable in polynomial time (leading to exact or very accurate gradients) is when the energy is parametrized with an ICNN. We believe our proposed loss is a clear advance in this setting."  It would be great to have this clarified in the paper.

---

> > > > > ### Author Response · Authors · 2022-08-09
> > > > > **Comments addressed, thank you**
> > > > >
> > > > > Thank you very much for the constructive comments. We hope that your concerns are now addressed satisfactorily.
> > > > >
> > > > > > I think the references should be discussed earlier in the intro
> > > > >
> > > > > This is now addressed in the revised manuscript. We now mention envelope theorems **before** introducing our contribution so it should be clear that we are not the first to use them. Instead, when introducing our contribution, we decided to mention regularization, which can be used to ensure that the unicity assumption of envelope theorems is satisfied. We note that papers [44] and [11] do not explicitly mention envelope theorems / Danskin's theorem, even though this is what they are doing (potentially without knowing it). We agree that the introduction is now indeed better, thanks!
> > > > >
> > > > > > but please just be precise about how different they are (in L565)
> > > > >
> > > > > This is now addressed on lines 570-573.
> > > > >
> > > > > > It would be great to have this clarified in the paper.
> > > > >
> > > > > This is now addressed on line 110.

---

> > > > > > ### Comment · Reviewer_jRDG · 2022-08-09
> > > > > > **I have raised my score**
> > > > > >
> > > > > > Thank you for making these amendments. I have now raised my score to 5 to account for this

---

> ### Author Response · Authors · 2022-08-06
> **Discussion with reviewer**
>
> We noticed that reviewer jRDG still has not acknowledged our rebuttal. We understand that this is the summer break but we would very much like to engage with the reviewer. We believe that our work strongly advances the field of energy-based models / energy networks and that the reviewer score is disproportionately low compared to our contributions (which we listed in our rebuttal). We stress once again that generalized conjugates are not a well-known tool at all in the ML community. We therefore believe that the claim of "lacking novelty" and "being straightforward" is not justified. We thank again the reviewer for their time.

---

### Official Review · Reviewer_w577 · 2022-07-24

**Rating:** 6
**Confidence:** 3
**Soundness:** 4 excellent
**Presentation:** 2 fair
**Contribution:** 3 good

**Summary:**

The main contribution of the paper is to introduce and study generalized Fenchel-Young losses, which are losses for regularized energy networks that mirror Fenchel-Young losses, but where the Euclidean inner product is replaced by the energy function \Phi. They compare their losses with existing losses in the settings of multilevel classification and imitation learning.

**Questions:**

- There are certain points which could be made clearer and should be expanded: (a) the paragraph in lines 140-148 (“Existing loss functions for energy networks”) is too condensed and hard to parse, it should be expanded, (b) section 6 is also hard to follow, it would be useful to explain at the beginning of the section what calibration is in this setting. On the other hand, sections 4 and 5 are reasonably clear.

- In section 6, Proposition 3, the list of properties is a bit long. Some of the properties are commented on below the proposition, but the attention of the reader gets lost, as it is not apparent which properties are most relevant. One suggestion is to defer to the appendix the properties that do not add much to the explanation.

- Computationally, how do generalized FY losses compare to the existing alternatives? It would be good to compare time.


**Limitations:**

Everything ok.

**Strengths And Weaknesses:**

Originality: The work can be considered a novel combination of existing techiques (Fenchel-Young losses and \Phi-conjugates). It is not strikingly original but still valuable. It is clear how the work differs from previous works, which are appropriately cited for the most part.

Quality: The work seems technically sound (I haven’t checked most proofs in the appendix).

Clarity: Some parts need to be clearer. See questions and suggestions.

Significance: The results provided are relevant as energy networks trained with generalized FY losses have better test accuracy in 4 of the 6 multilevel classification settings, and on 3 out of 4 tasks in imitation learning.

---

> ### Author Response · Authors · 2022-07-28
> **Comments taken into account (see revised manuscript)**
>
> Thank you for the positive assessment and constructive comments. We have taken into account your comments in the revised manuscript (modifications are highlighted in blue) and hope you will consider increasing your score.
>
> > The work can be considered a novel combination of existing techniques (Fenchel-Young losses and \Phi-conjugates). It is not strikingly original but still valuable.
>
> Technically, we do not combine Fenchel-Young losses but generalize them. We emphasize that, unlike classical conjugate functions (a.k.a. Legendre-Fenchel transforms), Phi-conjugate functions are not well-known, studied or used in the ML community. Therefore, we believe that our paper is original and will bring some awareness to this powerful tool.
>
> > the paragraph in lines 140-148 (“Existing loss functions for energy networks”) is too condensed and hard to parse, it should be expanded
>
> We agree and have added more details in the revised manuscript.
>
> > section 6 is also hard to follow, it would be useful to explain at the beginning of the section what calibration is in this setting
>
> Many times, notably for differentiability reasons, the loss used at training time (here, our generalized Fenchel-Young loss) is used as a surrogate / proxy for the loss that we use at test time (e.g., zero-one loss, precision at k, etc). Calibration guarantees ensure that we still minimize the excess of risk of the test loss (a.k.a. target loss), even though we use a different loss at train time. We added a clarifying paragraph.
>
> > In Proposition 3, the list of properties is a bit long. Some of the properties are commented on below the proposition, but the attention of the reader gets lost, as it is not apparent which properties are most relevant. One suggestion is to defer to the appendix the properties that do not add much to the explanation.
>
> Thank you for the feedback. We decided to keep the entire Proposition but have improved the connection between the Proposition and the explanations. Please check the revised manuscript.
>
> > Computationally, how do generalized FY losses compare to the existing alternatives? It would be good to compare time.
>
> In Table 5, we compared 4 loss functions : the proposed generalized FY loss, the cross-entropy loss, the generalized perceptron loss and the energy loss. The big-O complexity of the first 3 losses is dominated by the cost of solving the argmax problem (Equation 4 in the paper). Therefore their big-O complexity is the same. The energy loss is cheaper because it doesn’t require solving the argmax but it is known to perform poorly (LeCun 2006), as also confirmed by our empirical results in Table 5.

---

> > ### Comment · Reviewer_w577 · 2022-08-04
> > **Answer to revised manuscript**
> >
> > I acknowledge that the changes have been done and have no further comments.

---

### Author Response · Authors · 2022-07-28
**Revised manuscript**

We thank all reviewers for their comments, as well as the AC and senior AC for their editorial work.  We have posted a revised manuscript incorporating the reviewer comments (modifications are highlighted in blue color).

We have already done so to Reviewer jRDG but we would also like to recall our key (mathematical) contributions:
- Introducing the new notion of a regularized energy network.
- Using generalized conjugates, which are not well-known at all in the ML community. Our paper will bring awareness to this new tool.
- The smoothness result (Proposition 1, item 6) is new and not straightforward.
- The lower bound result (Proposition 3, item 5) is more general than the existing one for regular FY losses and uses a simpler proof.
- The calibration guarantees (Proposition 4) are more general than the existing ones for regular FY losses. They use a novel and not straightforward proof technique.
- Generalized Bregman divergences (Appendix A) are completely new and not straightforward.

Overall, we believe that our paper advances the field of energy networks by introducing a principled loss construction with theoretical guarantees.

We are happy to make further clarifications if needed.

---

### Meta-Review · Area_Chair_ykFj · 2022-08-25

**Recommendation:** Accept
**Confidence:** Certain

**Metareview:**

This paper introduces a new notion of regularized energy function using generalized Fenchel conjugates.  Reviewers were leaning towards accept, the least convinced reviewer discussed at length with the authors the contribution of the paper and the comparison of the proposed method to prior work, and leaned also towards accept after rebuttal and paper revision. Accept.

**Award:**

No

---

### Decision · Program_Chairs · 2022-09-14

Accept